# Detecting Distributional Treatment Responders with False Discovery Rate Control

## Abstract

In this paper, we introduce causal responder detection (CARD), a method for distributional responder analysis that identifies treated subjects whose outcomes significantly depart from the control response distribution while controlling the false discovery rate (FDR) marginally over the tested treated population. CARD builds on the AdaDetect framework and, in randomized settings, inherits finite-sample FDR control under the exchangeability conditions required by AdaDetect when coupled with the Benjamini–Hochberg procedure. For observational studies, we propose a propensity score–adjusted extension whose validity is asymptotic and depends on ignorability, overlap, and adequate propensity-score estimation. Simulation studies and real-data applications demonstrate that CARD effectively detects distributional treatment responders with high power across a range of heterogeneous distributional treatment-effect scenarios.

## 1 Introduction

Personalized medicine is expected to advance healthcare in the near future (Vicente et al., 2020). In contrast to a one-size-fits-all approach, personalized medicine advocates for treatments tailored to individual patients based on their clinical characteristics.

Responder analysis is a method of evaluating trial results by looking at the proportion of individual patients who achieved a predefined, clinically meaningful level of improvement, rather than just looking at the average improvement of the entire group (Henschke et al., 2014). This complements the traditional focus on average effects across all participants, which can obscure benefits concentrated in a particular subgroup (Guyatt et al., 1998). Responder analysis is widespread in clinical trials (Moore et al., 2010; Straube et al., 2010; Chuang et al., 2022). We focus on statistical evidence that a treated subject's outcome distribution deviates from the control distribution, rather than exceeding a predefined clinically meaningful threshold (Snapinn & Jiang, 2007). We refer to such subjects throughout as *distributional responders*: rejecting CARD's null does not certify that $Y_i(1) \neq Y_i(0)$ for unit $i$, since the counterfactual is unobserved and unit-level certification requires assumptions (e.g. monotonicity) beyond those used here. CARD also does not decompose the ATE or CATE into a proportion of responders: ATE and CATE summarize mean effects, whereas CARD targets distributional inconsistency with the control distribution and the two coincide only under additional structure.

Another approach to understanding heterogeneous treatment effects is estimating the conditional average treatment effect (CATE), which focuses on understanding how the effect of a treatment varies across different segments of the population, rather than only calculating the average treatment effect (ATE). Unlike ATE, which provides a single overall effect, CATE enables the identification of treatment benefits that may be specific to subgroups defined by their unique characteristics (Angus & Chang, 2021).

CATE and responder analysis complement each other. CATE offers insights into the heterogeneous treatment effects in terms of the expected values of certain population segments. Meanwhile, responder analysis provides additional context by indicating the proportion of total responders and extends beyond the expected value. For instance, consider a scenario where a certain treatment increases the variability within a subgroup of the population without necessarily affecting their expected value. Since CATE estimates only the conditional

expectation of the treatment effect, it is by construction unable to detect such distributional changes. By contrast, subjects in this subgroup could be identified as responders by a method sensitive to the full outcome distribution.

Individual treatment effects (ITE) provide a comprehensive view of the treatment response. Lei & Candès (2021) and Alaa et al. (2024) proposed methods for constructing confidence intervals for the ITE using split conformal prediction (Papadopoulos, 2008). Lei & Candès (2021) used weighted conformal prediction for combining the confidence intervals for the counterfactual to the ITE confidence interval. By contrast, Alaa et al. (2024) suggested a method based on inductive conformal prediction for specific types of meta-learners.

These confidence intervals offer more information than the CATE point estimate alone. When these confidence intervals are sufficiently powered they can be used for responder analysis, where subjects with confidence intervals that exclude 0 can be determined as responders. However, there are certain settings where they lack the statistical power to detect differences between treated and untreated responses (see Section 4). Duan et al. (2024) suggested an approach to test if the CATE of observations is non-negative (or positive) while controlling the false discovery rate (FDR) over all observations.

CARD (causal responder detection), our proposed responder analysis method, builds on AdaDetect (adaptive detection, Marandon et al. (2024)). AdaDetect is a conformal prediction method for detecting out-of-distribution samples using an adaptive non-conformity score. CARD offers two key innovations over AdaDetect for the task of responder identification. First, we suggest a specialized scorer aimed at discovering responders. This scorer uses recursive partitioning of the feature space, similar to the causal tree algorithm (Athey & Imbens, 2016), but differs by training a classifier to distinguish treated from untreated subjects based on $Y$ within each leaf, rather than estimating the CATE directly. This approach increases the power substantially compared to AdaDetect, which utilizes off-the-shelf classifiers as scorers. Second, we address the potential bias from non-random allocation of treatment by applying a propensity score based adjustment (Rosenbaum & Rubin, 1983; Tibshirani et al., 2019).

The suggested method tests whether each treated observation's $(Y_i, X_i)$ is sampled from the same distribution as the control response distribution, while controlling the FDR, as opposed to providing confidence intervals for individual observations as in Lei & Candès (2021); Alaa et al. (2024). Duan et al. (2024) also provides FDR guarantees, but targets a fundamentally different hypothesis: whether the CATE is non-negative. Because CARD tests for general distributional differences rather than directional mean effects, the two methods address distinct inferential targets, and their statistical power is not directly comparable. When the scientific question concerns the conditional mean, the approach of Duan et al. (2024) is more appropriate; when the interest lies in detecting any departure from the control distribution, CARD is the natural choice.

CARD provides finite-sample FDR control in randomized experiments and asymptotic FDR control in observational settings for identifying treated individuals whose observed outcomes are statistically inconsistent with the control outcome distribution. In contrast to existing FDR-controlled approaches that primarily focus on directional or mean-based hypotheses (e.g., testing whether the CATE is non-negative), CARD is designed to detect fully general distributional deviations, including changes in variance, skewness, or tail behavior.

The rest of Section 1 provides background on the inference task at hand. In Section 2 we give an overview of AdaDetect and causal trees, as they serve as building blocks for our method. In Section 3 we present CARD, discuss its components, and suggest an adjustment to handle cases where the treatment assignment depends on the covariates. In Section 4 we examine the performance of CARD in randomized controlled trials (RCTs) and observational study settings. Finally, we conclude with a discussion and suggested avenues for further research in Section 5.

## 1.1 Causal Inference

In this paper, we follow the potential outcomes framework (Imbens & Rubin, 2015) with a binary treatment. Given $n$ subjects, we denote by $T_i \in \{0, 1\}$, $(Y_i(1), Y_i(0))$ and $X_i$, the binary treatment indicator, potential outcomes and the vector of covariates (of length $p$) of subject $i$. Using brackets ($[\cdot]$) as a generic notation for a statistical distribution and the tilde ($\sim$) to mean "is distributed as," we assume that the potential

outcomes are distributed according to

$$(Y_i(1), Y_i(0), T_i, X_i) \sim [Y(1), Y(0), T, X].$$

Our focus will be on a binary treatment, $T_i \sim \text{Ber}(e(x))$, where $e(x)$ is the propensity score. Under the commonly assumed stable unit treatment value assumption (SUTVA), the observed data set comprises triplets $(Y_{\text{obs},i}, T_i, X_i)$ where $Y_{\text{obs},i} = T_i \cdot Y_i(1) + (1 - T_i) \cdot Y_i(0)$.

The ITE $\tau_i$ is defined as

$$\tau_i \equiv Y_i(1) - Y_i(0).$$

By definition, for each unit, only one potential outcome is observed while the other remains unobserved. Consequently, the ITEs are unobserved and must be inferred. We also assume strong ignorability throughout the paper:

$$(Y(1), Y(0)) \perp\!\!\!\perp T \mid X.$$

Strong ignorability assumes that there are no unmeasured confounders affecting both the treatment assignment and the potential outcomes. This assumption clearly holds in RCTs. Thus, under this assumption, the treatment assignment is essentially randomized (i.e., independent of potential outcomes) conditional on the covariate values. These assumptions identify the conditional distributions of each potential outcome:

$$[Y_i(1) \mid X = x] \sim [Y_i \mid X_i = x, T_i = 1],$$

$$\text{and } [Y_i(0) \mid X = x] \sim [Y_i \mid X_i = x, T_i = 0].$$

Consequently, mean contrasts such as the CATE are identified. By contrast, the joint conditional distribution $(Y(1), Y(0)) \mid X = x$ is not identified without additional assumptions, so neither the full distribution of $Y(1) - Y(0) \mid X = x$ nor unit-level treatment effects are identified in general. Typically, the interest lies in estimating the CATE,

$$\tau(x) = \mathbb{E}(Y(1) - Y(0) \mid X = x).$$

CATE estimation has garnered much attention in recent years, with multiple methods developed to estimate CATE using machine learning (Künzel et al., 2019; Wager & Athey, 2018; Nie & Wager, 2021). For example, the S-learner consists of training an outcome prediction algorithm $\hat{\mu}(X, T)$, which aims to predict $E(Y|X = x, T = t)$. The CATE estimate is then given by

$$\hat{\tau}(x) = \hat{\mu}(x, T = 1) - \hat{\mu}(x, T = 0).$$

Other parameters of the ITE distribution are also valuable targets. For example, Fort (2016) proposed estimating the conditional quantile treatment effect. CARD's distributional test is sensitive to shifts in any such parameter, not only the conditional mean, which motivates its use when treatment effects may manifest as changes in variance, skewness, or tail behavior.

## 1.2 Inferential Goal

The objective of distributional responder analysis is to flag treated units whose observed $(Y_i, X_i)$ is statistically inconsistent with the control response distribution. Inferring directly on $Y_i(1) - Y_i(0)$ would require additional structure. CARD avoids direct inference on $Y_i(1) - Y_i(0)$ by testing a distributional null. For each treated observation, the CARD null is that its observed outcome-covariate pair is consistent with the control counterfactual response law, with covariates distributed as in the treated population:

$$H_{0,i}^{\text{CARD}} : (X_i, Y_i(1)) \sim P_{X|T=1}(x) \, P_{Y(0)|X=x}(y).$$

Under ignorability, this null is identified from the observed control distribution as

$$H_{0,i}^{\text{CARD}} : (X_i, Y_i) \sim P_{X|T=1}(x) \, P_{Y|X=x, T=0}(y). \tag{1}$$

A natural alternative would be the conditional hypothesis $Y_i(1) \sim P_{Y|X=X_i, T=0}$, but conditional testing requires smoothness assumptions and increasing sample sizes (Foygel Barber et al., 2021). We therefore

control the FDR *marginally over the tested treated population*, not conditionally at a fixed covariate value $X = x$. As shown in Section 3.4 , conditional Type I error can in fact be arbitrarily large even while marginal control is preserved, so CARD discoveries should be read as FDR-controlled over the tested treated units as a population, not as conditionally valid statements for any particular covariate profile.

**Remark 1.** *CARD's null is a statement about distributions. If $Y(1)$ and $Y(0)$ are identically distributed (e.g. both $N(0, 1)$) but sampled independently, every unit has a nonzero ITE with probability one, yet the CARD null holds. Conversely, CARD can flag units whose treatment alters variance, skewness, or tail behavior with no change in conditional mean, scenarios invisible to CATE-based methods. Thus, a CARD rejection provides distributional evidence of response: the treated unit's observed outcome is unlikely under the control response distribution. This is distinct from directly certifying the unit-level counterfactual statement $Y_i(1) \neq Y_i(0)$.*

## 2 Related Work

In this section we provide the relevant background for CARD.

### 2.1 Causal Tree and Forest

Causal tree (CT) is a type of decision tree specifically designed for estimating CATE (Athey & Imbens, 2016). A tree is a partition, $\Pi$, of the covariate space of $X$. We denote by $l(x; \Pi)$ the partition $x \in l, l \in \Pi$. A partition is obtained by applying a partitioning algorithm, which is used to minimize a loss function on some sample $(y_i, x_i, t_i), i \in \mathcal{I}$. Denote by $\mathcal{I}_j = \{i : t_i = j\}$, $j = 0, 1$, and $n_j = |\mathcal{I}_j|$. Finally, denote $L(\mathcal{I}_j, x, \Pi) = \{i : i \in \mathcal{I}_j, x_i \in l(x; \Pi)\}$. In CTs, at each partition the CATE is estimated by,

$$\hat{\tau}(x_i; \Pi) = \frac{1}{|L(\mathcal{I}_1, x_i, \Pi)|} \sum_{v \in L(\mathcal{I}_1, x_i, \Pi)} y_v - \frac{1}{|L(\mathcal{I}_0, x_i, \Pi)|} \sum_{j \in L(\mathcal{I}_0, x_i, \Pi)} y_j . \tag{2}$$

The partition, $\Pi$ is obtained by minimizing the modified CATE mean square error loss,

$$Loss(\Pi, \mathcal{I}) = -\frac{1}{n} \sum_{i \in \mathcal{I}_0 \cup \mathcal{I}_1} \hat{\tau}^2(x_i; \Pi) + \frac{2}{n} \sum_{l \in \Pi} \left( \frac{n_0 \hat{V}(\{y_i : i \in \mathcal{I}_0, x_i \in l\})}{n} + \frac{n_1 \hat{V}(\{y_i : i \in \mathcal{I}_1, x_i \in l\})}{n} \right), \tag{3}$$

where $n = n_0 + n_1$ is the total number of observations, and $\hat{V}$ is the standard sample variance estimator. To tackle the selection bias in the leaf estimates (as the leaves are selected to maximize this estimate), the authors suggest splitting the data into a training and a test set. The training set is used to find the partitions that minimize Equation (3), and the test-set is used for estimating the CATE within each terminal node (Equation (2)).

The use of sample splitting ensures that the estimates of the CATE in the terminal nodes are unbiased. Thus, any standard inference approach can be used to test the difference in means between the treated and untreated subjects in each terminal leaf using the test set. The trade-off is that there are fewer observations for finding the optimal partition. Furthermore, the method is effective only when generating a single tree. When growing a forest, sample-splitting is not sufficient to ensure valid testing, and different methods for estimating the variance must be applied. These methods have only asymptotic guarantees (Wager & Athey, 2018), even in cases of RCTs.

## 2.2 Conformal Prediction

Conformal prediction is a framework that provides a method to generate a prediction set that contains the true target value with a $1 - \alpha$ confidence (Angelopoulos et al., 2023). Suppose that a practitioner wishes to learn $f$, a mapping from $X$ to $Y$, $f(X) = Y$. We focus on inductive conformal prediction in which the calibration set is independent of the samples used for training $\hat{f}$ (Papadopoulos, 2008).

Suppose there is a predictive model $\hat{f}$, trained on some $(X_i, Y_i)$, where $Y_i$ is a continuous response and $X_i$ are the covariates. Conformal prediction requires a specification of a non-conformity score, $s(x, y)$, (where larger values correspond to worse agreement between $x$ and $y$) and a calibration set $(X_i, Y_i)$, $i = 1, \ldots, k$. The non-conformity score is often learnt based on a training set. To test a new observation, the non-conformity score is computed for the calibration set, $s(\hat{f}(X_1), Y_1), \ldots, s(\hat{f}(X_k), Y_k)$. Assuming that a new observation is from the same distribution as the calibration set, then their non-conformity scores are also distributed identically, thus,

$$\mathbb{P}\left( s(X_{test}, Y_{test}) \geq q\left( (1 - \alpha)\frac{k + 1}{k} \right) \right) \leq \alpha,$$

where $q(p)$ is the $p^{th}$ empirical quantile based on the calibration set non-conformity scores. This can be used to test $H_0 : Y_{test}|X_{test} \sim Y_i|X_i$, or by inverting the acceptance region generating prediction-intervals (Shafer & Vovk, 2008; Haroush et al., 2021). The choice of the score function is critical for the performance of the conformal predictor. Romano et al. (2019) suggested conformal quantile regression (CQR), an adaptive score function based on quantile regression. This allowed for the conformal interval to vary as function of $x$, making it adaptive when facing heteroscedasticity. Romano et al. (2019) also generalized the approach and suggested conformal-intervals that are based on the conditional histogram of $Y|X$.

To test $H_0 : Y_{test} \sim P_{Y|X} \times P_X$, using the procedure described above, one must assume that $X_{test} \sim P_X$. When one is unwilling to make such assumption, weighted conformal prediction can be used. Tibshirani et al. (2019) suggested a modification to conformal prediction when facing covariate shift, where the score is weighted according to the likelihood ratio $w(x) = \frac{\mathrm{d}P_{test}(x)}{\mathrm{d}P_0(x)}$, where $P_0$ is the distribution of the calibration set covariates.

Lei & Candès (2021) proposed a method to obtain confidence intervals for the ITE. In the process, they demonstrated how to construct counterfactual confidence intervals. The method builds on the CQR and utilizes an estimation of the propensity score as weights in weighted conformal prediction to handle distribution shifts.

### 2.2.1 AdaDetect

Suppose there are two samples, a reference sample $\mathcal{I}_0 = \{i : X_i \sim P_0\}$, and a test sample $\mathcal{I}_1 = \{i : X_i \sim P_i\}$, where $P_i$ can either be $P_0$ or some other distribution. The interest lies in identifying the specific observations in the test sample who differ from the reference sample, i.e., testing,

$$H_{0,i} : X_i \sim P_0, \forall i \in \mathcal{I}_1. \tag{4}$$

Conformal prediction can be used for this purpose. First, a non-conformity score, $s$, which measures the degree of disagreement of an observation being sampled from $P_0$ is applied to $\mathcal{I}_0$ to estimate its empirical distribution under the null hypothesis (Haroush et al., 2021). Then, the same non-conformity is applied to each observation in $\mathcal{I}_1$. A p-value for each observation obtained according to $\mathrm{pv}_i = \frac{1 + \sum_{j \in \mathcal{I}_0} \mathbf{1}(s(x_j) \geq s(x_i))}{|\mathcal{I}_0| + 1}$, where $\mathbf{1}$ is the indicator function.

Bates et al. (2023) have shown that the resulting p-values adhere to a specific type of dependency and that the Benjamini-Hochberg (BH) multiplicity adjustment (Benjamini & Hochberg, 1995) can be safely applied to control the FDR. While the procedure is valid, it still remains to decide on $s$, as different scores will be powerful for different $P_1$ and $P_0$.

Recently, Marandon et al. (2024) proposed AdaDetect, an adaptive method for learning the non-conformity score $s$ for such hypotheses. Their non-conformity scores are the probabilities learned by a classifier aimed

at discriminating between the reference and test sets. Since the classifier is predicting on the same samples used for training, the probabilities are uncalibrated. Using these probabilities for testing Equation (4) will fail to control the FDR.

AdaDetect tackles this issue by employing "knockoffs," observations designed to be indistinguishable from actual test observations to the classifier, yet for which the null hypothesis (Equation (4)) holds. The non-conformity scores of the knockoffs are distributed similarly to the scores of true test observations under the null hypothesis, allowing estimation of the test-statistic distribution under the null.

The procedure involves sampling $k$ observations from $\mathcal{I}_0$ (denoted $\mathcal{I}_{ko}$) to serve as null calibration points.[1] These are combined with $\mathcal{I}_1$ to form $\mathcal{I}_1^* = \mathcal{I}_{ko} \cup \mathcal{I}_1$, while the remaining control observations form $\mathcal{I}_0^* = \mathcal{I}_0 \backslash \mathcal{I}_{ko}$.

The non-conformity score is learned by training a classifier aimed at discriminating between $\mathcal{I}_0^*$ and $\mathcal{I}_1^*$, where higher values indicate the sample is more likely to be from $\mathcal{I}_1^*$. The prediction will also serve as our non-conformity score. Intuitively, if $X_i \sim P_0$ then the $s(X_i), i \in \mathcal{I}_1$ are exchangeable with $s(X_j), j \in \mathcal{I}_{ko}$. Thus, a conformal p-value can be obtained as

$$\mathrm{pv}_i = \frac{1 + \sum_{j=1}^k \mathbf{1}\left(s(x_j) \geq s(x_i)\right)}{k+1} , \tag{5}$$

which quantifies how likely it is that the non-conformity score of a given observation is sampled from the non-responders' non-conformity score distribution. Due to the exchangeability of the knockoff scores and the tested observation score, each conformal p-value is marginally super-uniform under the null (i.e., $P(\mathrm{pv}_i \leq \alpha | H_0) \leq \alpha$). The use of a classifier generates an adaptive non-conformity score, which was shown to asymptotically achieve the optimal Neyman-Pearson test power (Marandon et al., 2024).

# 3 CARD

We propose CARD, a targeted approach that uses a specialized scorer to enhance the power in detecting changes in the response distribution (compared to the control) among treated subjects. Similar to AdaDetect, CARD employs a classifier to score observations of interest and uses knockoffs to control the FDR in finite samples. However, CARD differs by utilizing a scorer specifically tailored to the problem's structure, substantially improving its power compared to a direct application of AdaDetect with generic classifiers.

We initially discuss the suggested method in the RCT setting, which naturally avoids confounding due to differences in the distributions of $X|T = 1$ and $X|T = 0$ due to the randomization to treatment. In Section 3.3, we also discuss approaches for addressing such differences in non-randomized settings.

Let $\mathcal{I}_0$ and $\mathcal{I}_1$ denote the indices for untreated and treated observations, respectively. Additionally, define the set of treated observations indices where (Equation (1)) holds as $\mathcal{H}_0 = \{i \in \mathcal{I}_1 : (Y_i(1), X_i) \sim (Y_i(0), X_i)\}$.

## 3.1 Inference in RCTs

To test the hypotheses Equation (1), we split the untreated subjects into two groups, $\mathcal{I}_0 = \mathcal{I}_{ko} \cup \mathcal{I}_0^*$ and train a classifier to differentiate between $\mathcal{I}_1^* = \mathcal{I}_1 \cup \mathcal{I}_{ko}$ and $\mathcal{I}_0^*$. Let $s(x_i, y_i)$ be the resulting score according to the classifier. We then obtain p-values as in Equation (5),

$$\mathrm{pv}_i = \frac{1 + \sum_{j \in \mathcal{I}_{ko}} \mathbf{1}\left(s(x_j, y_j) > s(x_i, y_i)\right)}{k+1}.$$

In RCTs, randomization ensures that $X|T = 1 \sim X|T = 0$. If the null hypothesis (Equation (1)) is true for observation $i$, then $Y_i|T_i = 1, X_i \sim Y_i|T_i = 0, X_i$. Thus, $\{(X_i, Y_i)\}_{i \in \mathcal{I}_{ko}}$ are exchangeable with $\{(X_j, Y_j)\}_{j \in \mathcal{I}_1 \cap \mathcal{H}_0}$. If the classifier is invariant to the ordering of its training inputs (a property satisfied by

---

[1] We refer to these as "knockoffs" throughout, though this usage differs from the model-X knockoffs framework of Candes et al. (2018), where knockoffs are synthetic copies of covariates constructed for variable selection. Here, knockoffs are real control observations reassigned to the test set.

most practical classifiers; see eq. (8) in Marandon et al. (2024)), then by Lemma 3.2 and Theorem 3.3 of Marandon et al. (2024), the BH procedure applied to the resulting p-values will control the FDR.

To handle the setting of observational studies, where $X|T = 1 \not\sim X|T = 0$, we suggest a weighted version of AdaDetect based on weighted conformal prediction (Tibshirani et al., 2019); see Section 3.3. A graphical illustration of the procedure is provided in Figure 1.

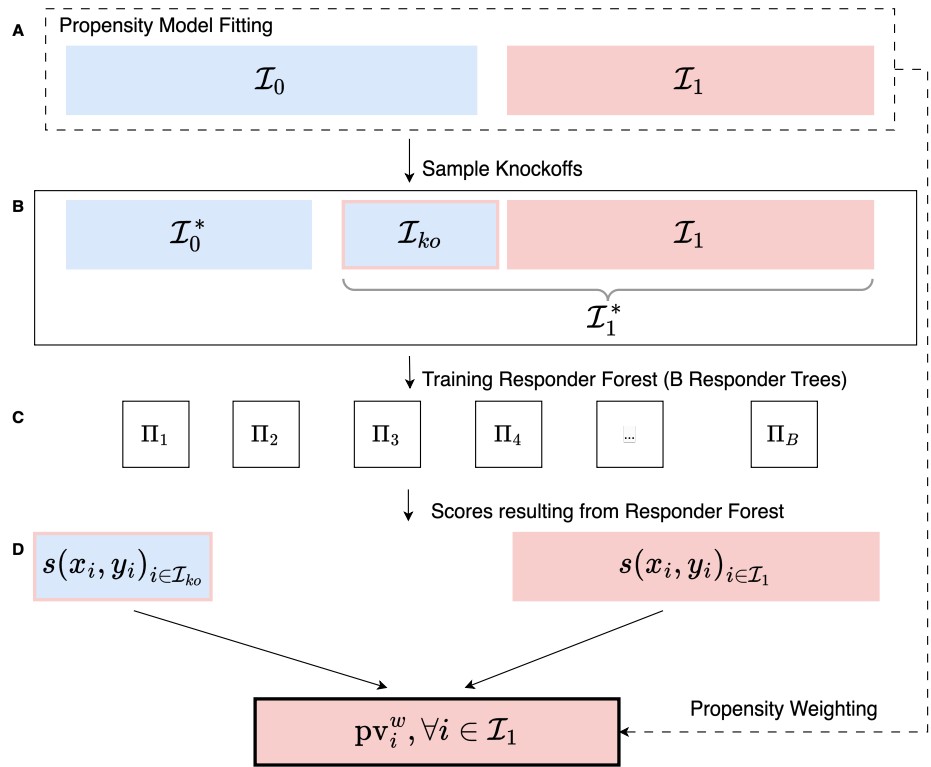

Figure 1: An illustration of CARD is provided where red fill indicates treated samples, and blue fill represents untreated samples. Knockoffs are depicted with a red frame. In step A, the propensity model is fit, and knockoffs are sampled from the untreated observations. Step B involves preparing the data for the classifier by combining the knockoffs with the treated observations. In step C, the responder forest is trained to classify between $\mathcal{I}_0^*$ and $\mathcal{I}_1^*$. Finally, in step D, the predictions from the responder forest are used as non-conformity scores. These scores are then weighted according to the propensity score to calculate p-values

## 3.2 Scorer

To increase power beyond that of AdaDetect, we suggest learning the non-conformity scores using a responder tree. The responder tree mirrors the principles of CTs, wherein the data is recursively partitioned on $X$ to minimize the overall $\text{Loss}_y$ across partitions until certain stopping criteria, such as reaching a maximum tree depth or convergence of loss are met. While the splits are done according to $X$ the evaluation in the terminal leaves is based solely on $Y$ and $T$.

The splits made by the responder tree are driven by the loss on the terminal nodes (where only $Y$ and $T$ are considered). The aim of the tree is to best classify observations by their treatment assignment (treated or not). The loss the responder tree attempts to minimize is

$$\text{Loss}(\Pi, \mathcal{I}_0, \mathcal{I}_1) = \sum_{l \in \Pi} c_l \, \text{Loss}_y(\{(y_i, t_i) : i \in \mathcal{I}_0 \cup \mathcal{I}_1, \ x_i \in l\}), \tag{6}$$

where $\text{Loss}_y$ is a classification loss (such as Gini impurity or cross-entropy) computed on the depth-2 tree that classifies $T$ from $Y$ within leaf $l$, and

$$c_l = \frac{|\{i \in \mathcal{I}_0 \cup \mathcal{I}_1 : x_i \in l\}|}{|\mathcal{I}_0 \cup \mathcal{I}_1|}.$$

We minimize the specified loss (Equation (6)) using recursive partitioning, a standard approach to optimize classification trees (Rokach & Maimon, 2005). The primary partitions are based on $X$. For each subset of observations defined by these partitions, we fit a shallow (depth 2) classification tree, where $Y$ is used to classify $T$. The loss associated with each partition is calculated based on the weighted classification loss of the shallow trees applied to each subset.

Intuitively, the responder tree focuses the classification task on detecting distributional discrepancies in $Y$ within covariate-adaptive regions of $X$. By partitioning on $X$ but evaluating splits using treatment-label predictability from $Y$, the scorer focuses on outcome shifts rather than generic covariate differences.

While a single responder tree is described, we can also consider a forest of scorers, where in a similar vein to Random Forest (RF), a bootstrap sample of observations and a sample of features are taken to grow each tree (Hastie et al., 2009). The score is the average prediction across the different trees as standard in classification random-forest.

### 3.3 Propensity Adjusted CARD

The propensity score is the probability of treatment assignment conditional on observed baseline characteristics, $e(x) = \mathbb{P}(T = 1 \mid X = x)$. Outside randomized experiments, $e(x)$ depends on $X$ and, when $Y(0) \not\perp X$, the knockoff observations are no longer exchangeable with the treated observations under the null. The propensity-adjusted procedure described below restores exchangeability *on average* and delivers FDR control *asymptotically*, but several caveats deserve emphasis before the construction:

- **The finite-sample RCT guarantee does not carry over.** CARD's finite-sample FDR control relies on the exchangeability of knockoffs and treated observations, which holds in RCTs by randomization. In observational settings, exchangeability is recovered only asymptotically and only when the propensity weights are correctly specified. We do not claim a finite-sample FDR guarantee in the observational data setting.

- **Ignorability and overlap are required.** The propensity-adjusted guarantee (Lemma 1) requires SUTVA, no unmeasured confounding, and strict positivity ($0 < e(x) < 1$ on the support of $X$). Violations of either ignorability or overlap can cause arbitrarily large biases that are not detected by the procedure itself.

- **Propensity-score estimation matters.** In practice $e(x)$ must be estimated. Misspecified propensity models, poor propensity distribution overlap, or overfit propensity scores degrade the empirical FDR control, particularly for smaller $n$. We characterize this empirically in Appendix H by varying overlap strength and propensity-model specification.

- **Error control is marginal, not conditional.** As discussed in Section 3.4 below, the observational FDR guarantee, like its randomized counterpart, is marginal over the tested treated population, and not conditional on $X = x$.

To address the bias from non-random treatment assignment, we use the propensity-adjusted p-value

$$\text{pv}_i^w = w_i^* + \sum_{j \in \mathcal{I}_{ko}} w_j^* \cdot \mathbf{1}\left(s(x_j, y_j) > s(x_i, y_i)\right), \tag{7}$$

with weights $w_j = e(x_j)/(1 - e(x_j))$, $j \in \{i\} \cup \mathcal{I}_{ko}$ and $w_j^* = w_j / \sum_{l \in \{i\} \cup \mathcal{I}_{ko}} w_l$. This is similar to the weighting used in (Lei & Candès, 2021).

**Lemma 1.** *Under the null hypothesis (Equation (1)), and assumptions A1-3, the weighted p-values (7)
control the Type I error at level $\alpha$,*

$$\mathbb{P}(\mathrm{pv}_i^w < \alpha) \leq \alpha.$$

*A.1.* **SUTVA (Stable Unit Treatment Value Assumption)**: *The observed outcome equals the potential outcome under the received treatment, i.e.,*

$$Y_{obs,i} = T_i \cdot Y_i(1) + (1 - T_i) \cdot Y_i(0).$$

*A.2.* **Ignorability (Unconfoundedness)**: *Conditional on covariates $X$, treatment assignment is independent of the potential outcomes,*

$$(Y_i(0), Y_i(1)) \perp\!\!\!\perp T_i \mid X_i.$$

*A.3.* **Overlap (Positivity)**: *Each treatment level is possible for all values of $X$,*

$$0 < \mathbb{P}(T_i = t \mid X_i = x) < 1 \quad \text{for all } t \in \{0,1\}, \text{ and for all } x \text{ such that } \mathbb{P}(X_i = x) > 0.$$

The proof is given in Appendix C. The lemma is stated for oracle propensity scores. When $e(x)$ is estimated, the result should be interpreted as asymptotic and contingent on adequate nuisance estimation. The simulations in Section 4.2 and Appendix H evaluate the finite-sample consequences of propensity estimation error. It is important to note that the resulting p-values do not have a theoretical guarantee for controlling the FDR in finite samples, particularly in the context of non-randomized experiments. In our experiments, the weighted p-values often control FDR in moderate settings, but finite-sample performance depends on overlap and propensity estimation quality (see Appendix H). Furthermore, it is shown that as $n \to \infty$, the BH procedure using the weighted p-values recovers the theoretical control of the FDR (Jin & Candès, 2023).

Although results are typically presented with oracle propensity, in practice, the propensity score is estimated from the same data we intend to analyze. To reduce over-fitting, we fit the propensity estimator using $m$-fold fitting (Jacob, 2020), training on $m-1$ folds and estimating the propensity on the remaining fold. We examine the empirical validity of this adjustment in Section 4.2.

### 3.4 Marginal vs. conditional error control

CARD tests the joint null hypothesis in (1), namely that a treated observation's pair $(X_i, Y_i)$ is distributed as a draw from the control response law with covariates distributed as in the treated population. Consequently, the resulting conformal p-values are super-uniform only marginally over the distribution of covariates in the tested treated population:

$$\mathbb{P}_{(X_0, Y_0) \mid T=1}(\mathrm{PV}_i \leq \alpha) = \\ \int \mathbb{P}_{X \mid T=1}(X = x) \mathbb{P}_{Y(0) \mid X=x}(\mathrm{PV}_i \leq \alpha)) \, dx \leq \alpha. \tag{8}$$

This guarantee does not imply that $\mathbb{P}(\mathrm{pv}_i \leq \alpha \mid X = x) \leq \alpha$ for every fixed covariate value $x$ when the null-hypothesis holds.

This distinction can be important. Consider, for example, $X \sim U[0,1]$, $T \sim \mathrm{Ber}(0.5)$, $Y(0) \mid X = \epsilon$, and $Y(1) \mid X = \mathbf{1}\{X > 0.01\} + \epsilon$, with $\epsilon \sim N(0, 10^{-7})$. For large samples, a flexible discriminator can learn to split first on $X > 0.01$ and then on a threshold in $Y$, such as $Y > 10^{-5}$. In this construction, observations with $X < 0.01$ satisfy the conditional null $Y(1) \mid X = x \sim Y(0) \mid X = x$, yet the learned joint scorer may still reject many or all such observations. This does not violate CARD's guarantee, because the guarantee is averaged over $X \mid T = 1$ rather than holding pointwise in $x$.

The practical implication is that CARD discoveries should be interpreted as FDR-controlled over the tested treated population as a whole, not as conditionally valid certifications for every patient profile or covariate stratum. In particular, the marginal guarantee does not bound the false discovery proportion within an arbitrary covariate subgroup. Achieving conditional FDR control would require additional assumptions, such as smoothness, and sample sizes that grow with the desired conditioning resolution (Foygel Barber et al., 2021).

## 4 Simulations

### 4.1 RCT Simulations

We begin by evaluating the performance of CARD in RCT settings using a simulation study. We use a variant of the data generation process that was used by Lei & Candès (2021) and Wager & Athey (2018). The data generation process is outlined as follows: We sample $X^* \sim N(0, \Sigma_{p \times p})$, where $\Sigma_{i,j} = \rho^{|i-j|}$. To obtain $X$ we transform $X_i = \Phi(X_i^*)$. The potential outcomes are generated according to,

$$Y(0) = 4 \cdot (X_1 + X_2) + \epsilon,$$
$$Y(1) = 4 \cdot (X_1 + X_2) + r f(X_1) \cdot f(X_2) + \epsilon,$$

where $\epsilon \sim N(0, \sigma(x)^2)$ and

$$f(x) = \mathbf{1}(x > 0.5) \cdot \frac{2}{1 + \exp(-12 \cdot (x - 0.5))}.$$

That is, the null hypothesis (Equation (1)) is false for $X_1 > 0.5, X_2 > 0.5$. Treatment assignments are randomly determined following a Bernoulli distribution, Ber(0.5). We explore several variations in the parameters: $r \in \{-1, +1\}$ to assess the effect of signal sign, $\sigma(x) \in \{1, -2 \cdot \ln(x)\}$ to compare homoscedastic and heteroscedastic settings, $\rho \in \{0, 0.9\}$ to distinguish between independent and dependent settings, and $p \in \{10, 100\}$ to evaluate the impact of the number of features. We also vary the number of observations, $n \in \{250, 500, 1000, 2000\}$. Each setting is repeated 100 times, and the averages are reported.

We compare CARD with several other methods of detecting distributional responders. Unless stated otherwise, CARD uses a responder forest with 50 trees, maximum depth 6, 20% of untreated observations reserved as knockoffs. Propensity scores in observational experiments are estimated using cross-fitting. Additional sensitivity to forest size, tree depth, and knockoff proportion is reported in Appendix G.

1. Global Testing: testing whether the treated outcome ($Y_i \mid T_i = 1$) is consistent with the control distribution $Y(0)$, which is also valid for testing Equation (1). The p-value is obtained on the basis of the $Y(0)$ eCDF (see Appendix B).

2. AdaDetect (Marandon et al., 2024): we apply the original AdaDetect method with a random-forest classifier on the joint $(X, Y)$ features, with the scikit-learn default `RandomForestClassifier` settings (100 trees, no depth limit). Again, 20% of the untreated observations are used as knockoffs.

3. CQR (Romano et al., 2019): we apply the CQR non-conformity score to obtain p-values (see Appendix A), equivalent to the counterfactual confidence intervals of Lei & Candès (2021). The two conditional quantiles at levels 0.025 and 0.975 are estimated by paired gradient-boosting regressors with default scikit-learn settings, using grid-searched quantile-level adjustment with 10 candidates per side. The untreated sample is split 80% for estimation of the quantile regressors and 20% for calibration.

In the RCT setting considered here, each method yields valid p-values for testing departures from the relevant control response distribution, and we apply BH to each set of p-values. The FDR is expected to be controlled at level 0.1. Other CATE estimation methods can also be used for testing for responders such as CF, BART and bootstrap with Meta-Learners (Wager & Athey, 2018; Chipman et al., 2012; Künzel et al., 2019). However, Lei & Candès (2021) showed they lack finite coverage control, and thus we omit them from formal comparisons.

The power results are presented in Figure 2, with additional FDR and power analyses provided in Appendix D. The use of AdaDetect with RF as the classifier yields poor results, which is unsurprising as RF attempts to detect differences across all features, unlike competing methods focusing on $Y$ or $Y|X$.

CARD is most advantageous when the distributional responder signal is heterogeneous, covariate-dependent, or not visible from the marginal distribution of $Y$. In simpler regimes, especially homoscedastic positive-signal settings at smaller sample sizes, global testing or CQR can be more efficient because they avoid the cost of learning a covariate-adaptive scorer.

As implied by its name, the global method performs well when responders are discernible solely by considering $Y$, particularly in scenarios with positive signal sign and homoscedastic noise. Because the global method does not split the data, it significantly outperforms other methods in this specific scenario. However, when the differences in distributions are only discernible upon conditioning on $X$ (see Appendix B for an illustrative example), the method has virtually no power.

Unlike CARD, the CQR non-conformity score is learnt only on the basis of the untreated distribution. This leads to much better performance of CARD across all settings except for positive signal and homoscedastic noise.

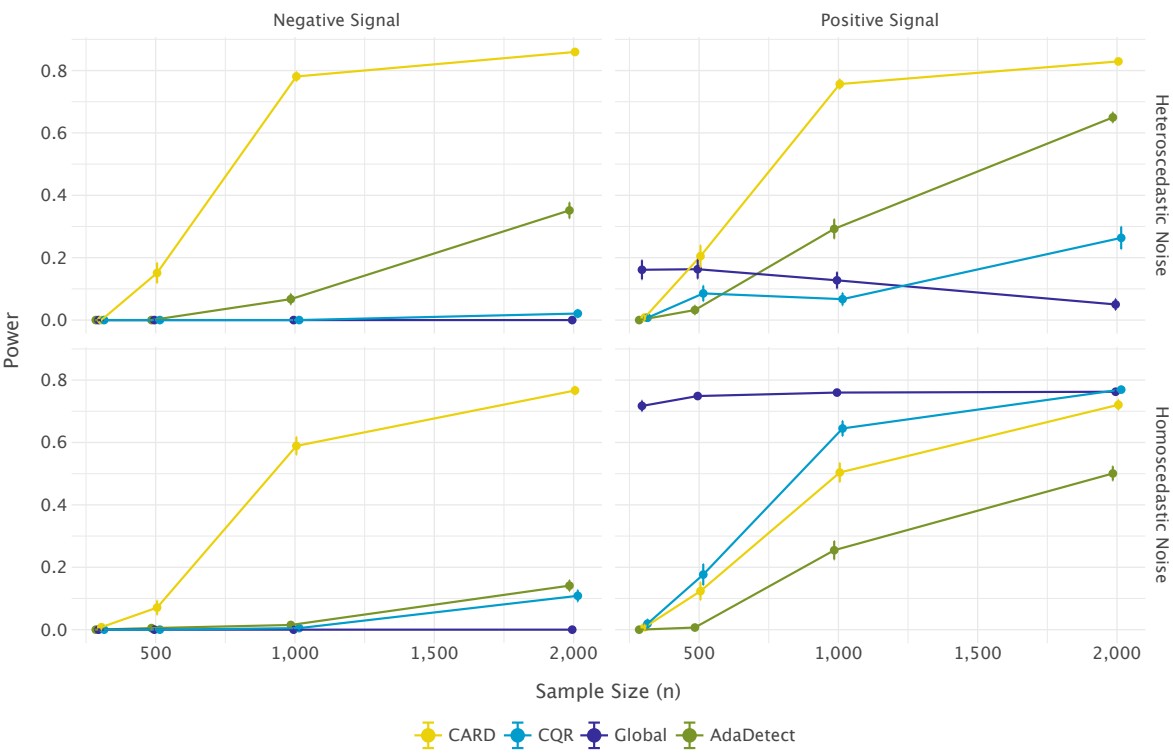

Figure 2: Power analysis of various methods for different sample sizes. The plots display the power as a function of sample size (n) in the $p = 10$ scenario. The power of CARD is higher in all scenarios except for the homoscedastic noise and positive signal, where it reaches parity when $n$ is sufficiently large. Intervals indicate $\pm 1$ s.e. around the estimate.

## 4.2 Observational Study Simulation

The asymptotic control of the FDR in the weighted CARD procedure is theoretically assured. In the following section, we examine the impact of weighting on CARD in finite sample settings.

We employ the same data generation protocol as outlined in Section 4, changing only the propensity function to mimic an observational study with confounders, $e = \frac{1}{1+\exp(1.75 \cdot X_1 - 0.825)}$. The propensity function ensures that the probability of receiving treatment is 0.5. We present results for the $p = 10$ and positive signal scenario, as the results are consistent across scenarios (see Appendix D). Furthermore, we restrict our simulations to CARD.

We consider four methods of handling non-trivial propensity: 1) None, ignoring propensity and applying CARD without any adjustment; 2) oracle adjustment, where weighting is based on true propensity; 3) RF classifier propensity adjustment; and 4) logistic regression propensity adjustment. The propensity models are fit using 10-fold cross-fitting.

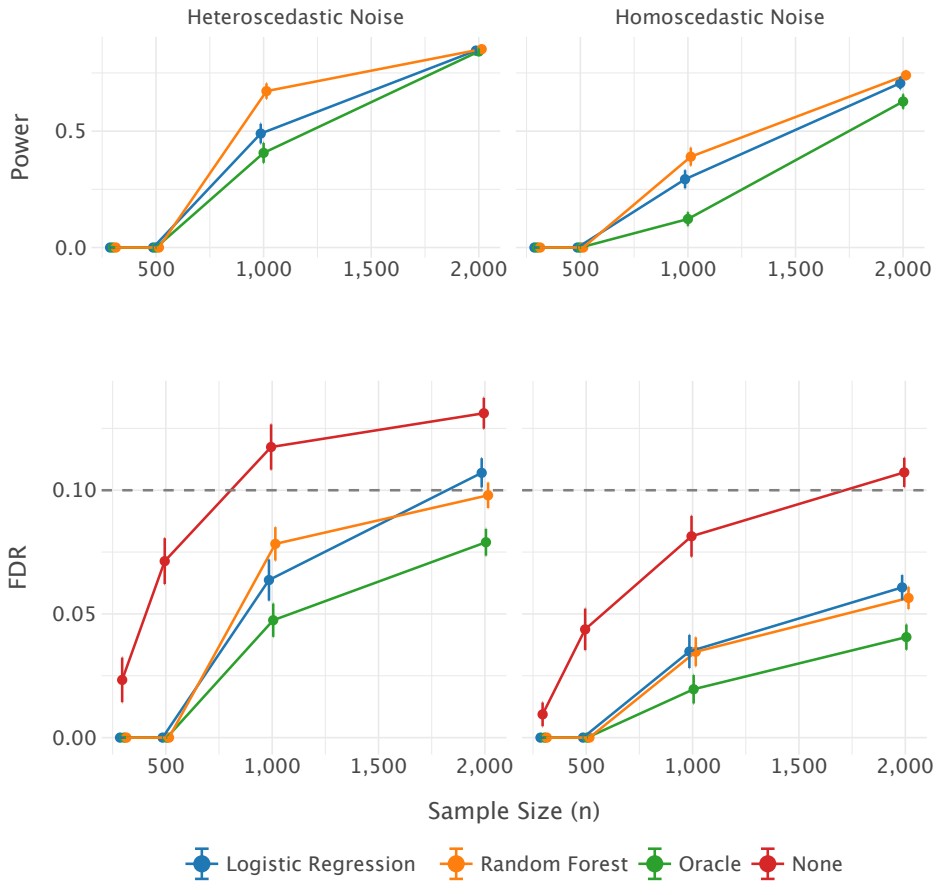

Figure 3: Power and FDR of the various methods of handling propensity. The None method is removed from the power (upper row) plots, since it fails to control the FDR at the expected level (bottom row). Intervals indicate ±1 s.e around the estimate.

The results, shown in Figure 3, indicate that all methods except "None" control the FDR at the expected level of 0.1. The estimating methods exhibit higher power than the oracle, a result attributed to cross-fitting and errors in propensity estimation. However, since the method is generally conservative, it controls the FDR at the expected level, which is promising for real-world applications. An example analysis using a subset of the ACIC 2016 data (Dorie et al., 2019) is presented in Appendix E. An additional real-world data example is provided in the next subsection.

### 4.3 Exploratory Real-World Data Example: Semaglutide in Obstructive Sleep Apnea

We also apply CARD to a real-world observational cohort of patients diagnosed with obstructive sleep apnea (OSA); full details are provided in Appendix F. The analysis is intended as an exploratory example of CARD in a high-dimensional, confounded setting, rather than as a confirmatory analysis of semaglutide effectiveness. The cohort includes $23,999$ patients with complete baseline covariates. Patients were followed for two years: a one-year screening period followed by a one-year weight-loss tracking period. Treatment was defined as initiation of semaglutide, and the control group included patients receiving no pharmacological weight-loss intervention. For untreated patients, a weight-loss start date was assigned at random to avoid immortal time bias. The outcome was relative weight change during the tracking period.

Since treatment assignment is non-random, we used propensity-adjusted CARD. The propensity score $e(x) = \mathbb{P}(T = 1 \mid X = x)$ was estimated using a random forest classifier with 5-fold cross-fitting. As shown in Table 2

and discussed in Appendix F, the propensity model had ROC AUC 0.851, indicating substantial selection into treatment. The common support was also limited, with 51.3% of patients having estimated propensity scores in $[0.05, 0.95]$. Thus, this example represents a challenging observational setting with strong confounding and imperfect overlap.

Using propensity-adjusted CARD with BH correction at $\alpha = 0.1$ yielded no formal FDR-controlled discoveries. We therefore do not interpret the analysis as identifying confirmed distributional responders. To examine whether the CARD scores nevertheless captured treatment-related signal, we also considered nominal, unadjusted CARD p-values at threshold 0.1. Under this exploratory threshold, CARD identified 585 of 2,530 treated patients as responders (23.1%; Table 2). These patients had mean relative weight change of $-8.04\%$, compared with $-2.67\%$ among the remaining treated patients, corresponding to a standardized mean difference of $-0.931$.

As a placebo diagnostic, we applied the same procedure to a randomly partitioned untreated population. This yielded a nominal discovery rate of 3.1%, compared with 23.1% in the semaglutide-treated group. Although this does not provide formal FDR control for the exploratory nominal discoveries, it suggests that the CARD scores capture treatment-associated signal rather than only procedural noise. Finally, a secondary random forest analysis separating nominal responders from non-responders identified prior anti-diabetic medication use, age at enrollment, and baseline year as the main predictors (see Appendix F and Table 2). These results should be interpreted as hypothesis-generating, but they illustrate the behavior of CARD in a realistic setting where strong confounding and limited overlap reduce the power of the formal procedure.

## 5  Discussion

CARD detects distributional responders with finite-sample FDR control in randomized experiments. In observational studies, the propensity-adjusted extension has an asymptotic validity argument under ignorability, overlap, and correctly specified or consistently estimated propensity scores, but its finite-sample behavior can degrade under poor overlap or misspecification (Appendix H). The overlap-sensitivity experiments also show that regularized propensity estimators can preserve power under weak overlap at the cost of residual bias, and in sufficiently misspecified interaction settings empirical FDR can exceed the nominal level. The detection of distributional responders complements CATE analysis: while CATE measures average treatment effects within population segments, CARD targets treated units whose outcome distributions are inconsistent with the control distribution, including shifts in variance, skewness, or tails that can be missed by mean-based methods.

**When CARD is not the right tool.** CARD is designed for regimes in which the treatment effect is heterogeneous or covariate-dependent and may manifest beyond the conditional mean. When the treatment effect is already visible in the marginal distribution of $Y$ (the homoscedastic positive-signal small-$n$ regime is an example such setting) simpler procedures that avoid learning a covariate-adaptive scorer, such as global testing on the eCDF of $Y(0)$ can outperform CARD, particularly for $n < 2000$. This is consistent with CARD's design: the gain from a responder-specific scorer is realized when the marginal distribution alone is uninformative, and the cost is paid up front in the form of a learned scorer that may be unnecessary when the signal is simple. Section 4.1 and Appendix G report these regimes, and we recommend reporting both CARD and a marginal baseline in applied settings where the operating regime is unknown.

**Computational cost.** The dominant computational cost of CARD is fitting the responder tree or forest on the augmented treated-plus-knockoff sample; conformal scoring and BH adjustment are minor post-processing steps. With a tree- or forest-based scorer, CARD's wall-clock cost is comparable to applying AdaDetect with the same classifier family, or to fitting a standard random-forest classifier of similar capacity on the same sample. We do not report implementation-dependent wall-clock numbers; runtime instrumentation is recorded alongside every simulation in the released code and may be inspected directly.

**Marginal versus conditional error control.** As discussed in Section 3.4, CARD controls error *marginally* over the tested treated population, not conditionally on $X = x$. This is shared with counterfactual ITE confidence intervals (Lei & Candès, 2021) and reflects a fundamental limit on conditional

inference under finite samples (Foygel Barber et al., 2021). Extending CARD to conditional hypotheses, under explicit smoothness assumptions, remains an interesting direction.

While CARD is presented as a method for detecting distributional responders, it can also be applied to other use-cases:

1. A global null test for treatment effectiveness, allowing for quick screening of various datasets.

2. Identifying subspaces of $X$ in which the treatment and non-treatment response distributions differ. A single responder tree tries to find a subspace of $X$ where the distributions $Y(0)|X \in A$ and $Y(1)|X \in A$ are different. Testing the difference between the two distributions using only the knockoffs in the identified subspace, $\mathcal{I}_{ko} \cap A$ provides a valid test for distribution differences given $A$.

Similar to Duan et al. (2024), CARD can also be configured to test only for positive (or negative) responders. Another issue with CARD (and any other split-conformal prediction method) is its inherent randomness due to the split (or selection of knockoffs). Fortunately, by pairing CARD with the method suggested by Meinshausen & Bühlmann (2010), most of this randomness can be removed while maintaining similar power, we investigate the stability of the procedure and how it can be improved in Appendix I. Finally, combining this method with various techniques for constructing valid confidence intervals for the true discovery proportion (Hemerik & Goeman, 2018; Millstein et al., 2022) can offer further insights into the proportion of responders.

## Broader Impact Statement

CARD is intended as a statistical tool for identifying distributional responders to a treatment, not as a decision rule for treatment allocation. We caution against using CARD outputs as a sole basis for assigning or withholding treatment for several reasons. First, the FDR guarantee is marginal: CARD does not produce conditionally valid statements for any particular patient or covariate stratum, and naive subgroup-level use is not supported by the theory (Section 3.4). Second, in observational settings, unmeasured confounding, overlap violations, and biased propensity estimation can produce misleading discoveries or unequal performance across subgroups; populations with poor overlap are especially vulnerable to both false discoveries and missed real effects (see Appendix H). Third, CARD's exploratory use on real-world data, as illustrated by the semaglutide example, is intended to flag candidate subpopulations for follow-up rather than to ground individual treatment decisions. We recommend that any deployment be accompanied by overlap diagnostics, propensity sensitivity analyses, derandomization for stability, and domain oversight, and that CARD findings be validated through confirmatory designs before influencing clinical practice.

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

## A   Conformal Quantile Regression (CQR)

To produce CQR, we split the data into a training set, $\mathcal{I}_{tr}$, and a calibration set, $\mathcal{I}_{cal}$. First, the training set is used for fitting a quantile regression model (such as linear quantile regression, quantile boosting, quantile trees, etc.). The model is trained to predict the conditional quantiles $q_{\alpha_l}(x)$ and $q_{\alpha_h}(x)$, where $\alpha_l + \alpha_h = \alpha$, and $1 - \alpha$ is the coverage probability. Then, the calibration set is used to estimate the non-conformity score,

$$s(X_i, Y_i) = \max\left(q_{\alpha_l}(X_i) - Y_i, Y_i - q_{\alpha_h}(X_i)\right).$$

To construct the CI, the $1 - \alpha$ quantile of the non-conformity scores are computed on the calibration set, denoted by $Q_{1-\alpha}$. Finally, the confidence interval is,

$$C(X_{n+1}) = [q_{\alpha_l}(X_i) - Q_{1-\alpha}, q_{\alpha_h}(X_i) + Q_{1-\alpha}].$$

To obtain p-values, we use the following,

$$\text{pv}_i = \frac{1 + \sum_{x_j, y_j \in \mathcal{I}_{cal}} \mathbf{1}\left(s(x_j, y_j) \geq s(x_i, y_i)\right)}{|\mathcal{I}_{cal}| + 1}. \tag{9}$$

## B   Global Testing

The task of detecting responders is essentially a task of detecting out-of-distribution observation based on the response. A simple approach would be to calculate the empirical p-values of $Y_i, T_i = 1$ observations based on the null sample, the p-values are,

$$\text{pv}_i^{\text{global}} = 2 \min\left(\frac{1 + \sum_{j=1}^{n_0} \mathbf{1}\left(y_j \geq y_i\right)}{n_0 + 1}, \frac{1 + \sum_{j=1}^{n_0} \mathbf{1}\left(y_j \leq y_i\right)}{n_0 + 1}\right).$$

A major drawback of the method is its inability to handle subjects which are responders conditional on some $X$. To illustrate the issue, we will use a somewhat contrived example, which exaggerates what otherwise is a very common phenomena. Suppose that $X \sim U[0, 1]$,

$$Y(0) = -5 \cdot \mathbf{1}(x < 0.5) + 5 \cdot \mathbf{1}(x > 0.5) + \epsilon,$$

and

$$Y(1) = -5 \cdot \mathbf{1}(x < 0.25) + 5 \cdot \mathbf{1}(x > 0.5) + \epsilon,$$

where $\epsilon \sim N(0, 1)$.

The global testing approach has no power, since the responder do not appear extreme considering $Y$ (see Figure 4 A), while when considering $0.25 < X < 0.75$, it is clear that the observation centered around $Y = 0$ are anomalies (see Figure 4 B). Indeed, in this example the power of the CARD is 1, while the global testing has a power of 0.

## C   Proof of the Lemma in Lemma 1

*Proof.* For simplicity, we assume that $\mathcal{I}_{ko} = \{1, \dots, k\}$ and $i = k + 1$.

Denote $Z_i = (X_i, Y_i)$, $z_i = (x_i, y_i)$. Let $c = \frac{\mathbb{P}(T=0)}{\mathbb{P}(T=1)}$, and let

$$\tilde{w}_i(x, y) = \begin{cases} 1 & 1 \leq i \leq k \\ c\frac{e(x)}{1 - e(x)} & i = k + 1 \end{cases}.$$

Under the null hypothesis, it holds that for every $x, y$, $1 \leq i_0 \leq k$

$$\mathbb{P}((X, Y) = (x, y)|T = 0)\tilde{w}_{k+1}(x, y) = \mathbb{P}((X, Y) = (x, y)|T = 1)\tilde{w}_{i_0}(x, y). \tag{10}$$

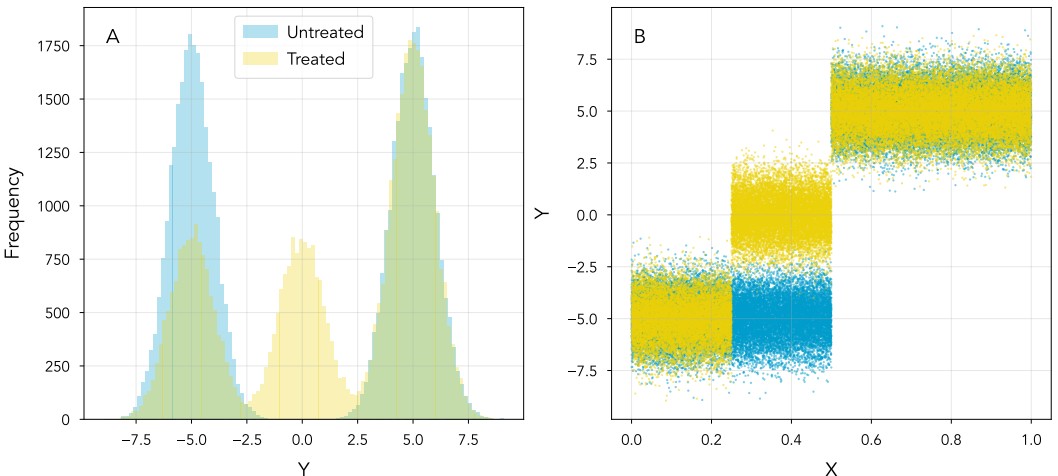

Figure 4: A. Histogram of $Y$, the responders are located between the two modalities of $Y(0)$. B. Scatter plot of $Y$ $X$, it is clear to see the responders around $Y = 0$ and $0.25 < X < 0.5$.

Indeed, the left hand side of Equation (10) is

$$\frac{\mathbb{P}((X,Y) = (x,y)|T=0)\mathbb{P}(T=0)P(T=1|X=x)}{\mathbb{P}(T=1)P(T=0|X=x)} =$$

$$\frac{\mathbb{P}(T=0|(X,Y)=(x,y))\mathbb{P}((X,Y)=(x,y))\mathbb{P}(T=1|X=x)}{\mathbb{P}(T=1)\mathbb{P}(T=0|X=x)} =$$

$$\frac{\mathbb{P}((X,Y)=(x,y))\mathbb{P}(T=1|X=x)}{\mathbb{P}(T=1)} =$$

$$\mathbb{P}((X,Y) = (x,y)|T=1).$$

The first equality is by Bayes rule, the second equality is due to the null hypotheses ($\mathbb{P}(Y = y, T = t|X = x) = \mathbb{P}(Y = y|X = x)\mathbb{P}(T = t|X = x)$) and the last equality is by definition.

Equation (10) implies that $(Z_1, ..., Z_{k+1})$ are weighted exchangeable ((Tibshirani et al., 2019, Definition 1)) – for every permutation $\sigma$,

$$\frac{\mathbb{P}(Z_1 = v_1, \ldots, Z_{k+1} = v_{k+1})}{\tilde{w}_1(v_1) \cdots \tilde{w}_{k+1}(v_{k+1})} = \frac{\mathbb{P}(Z_1 = v_{\sigma(1)}, \ldots, Z_{k+1} = v_{\sigma(k+1)})}{\tilde{w}_1(v_{\sigma(1)}) \cdots \tilde{w}_{k+1}(v_{\sigma(k+1)})}.$$

Now, we let $E_z$ be the event that $\{Z_1, ..., Z_{k+1}\} = \{z_1, ..., z_{k+1}\}$. By the above, there is a constant $C_z$ such that

$$\mathbb{P}(Z_1 = v_{\sigma(1)}, \ldots, Z_{k+1} v_{\sigma(k+1)}) = C_z \tilde{w}_1(v_1) \ldots \tilde{w}_{k+1}(v_{k+1})$$
$$= C_z \tilde{w}_{k+1}(v_{\sigma(k+1)}).$$

Assuming for simplicity that all the values are different, using the same steps as in the proof of (Tibshirani et al., 2019, Lemma 3), it holds that

$$\mathbb{P}(Z_{k+1} = z_j|E_z) = \frac{\sum_{\sigma:\sigma(k+1)=j} \mathbb{P}(Z_1 = v_{\sigma(1)}, \ldots, Z_{k+1} = v_{\sigma(k+1)})}{\sum_\sigma \mathbb{P}(Z_1 = v_{\sigma(1)}, \ldots, Z_{k+1} = v_{\sigma(k+1)})}$$
$$= \frac{k!\tilde{w}_{k+1}(z_j)}{\sum_{l=1}^{k+1} k!\tilde{w}_{k+1}(z_l)} = w_j^*.$$

Finally, notice that the score $s$ is constant given $E_z$. This implies together with the above

$$\mathbb{P}(\text{pv}_i^w < \alpha | E_z) \le \alpha.$$

And after marginalizing we get the required result. □

# D   Additional RCT Simulations

## D.1   Power Results

Here we offer additional power results for the RCT setting simulations (Section 4.1). The plots shown in Figure 5 illustrate the power as a function of the number of observations. The conclusions from Section 4.1 are similar across the various scenarios. The CARD method outperforms all methods except in the positive signal and homoscedastic noise scenario.

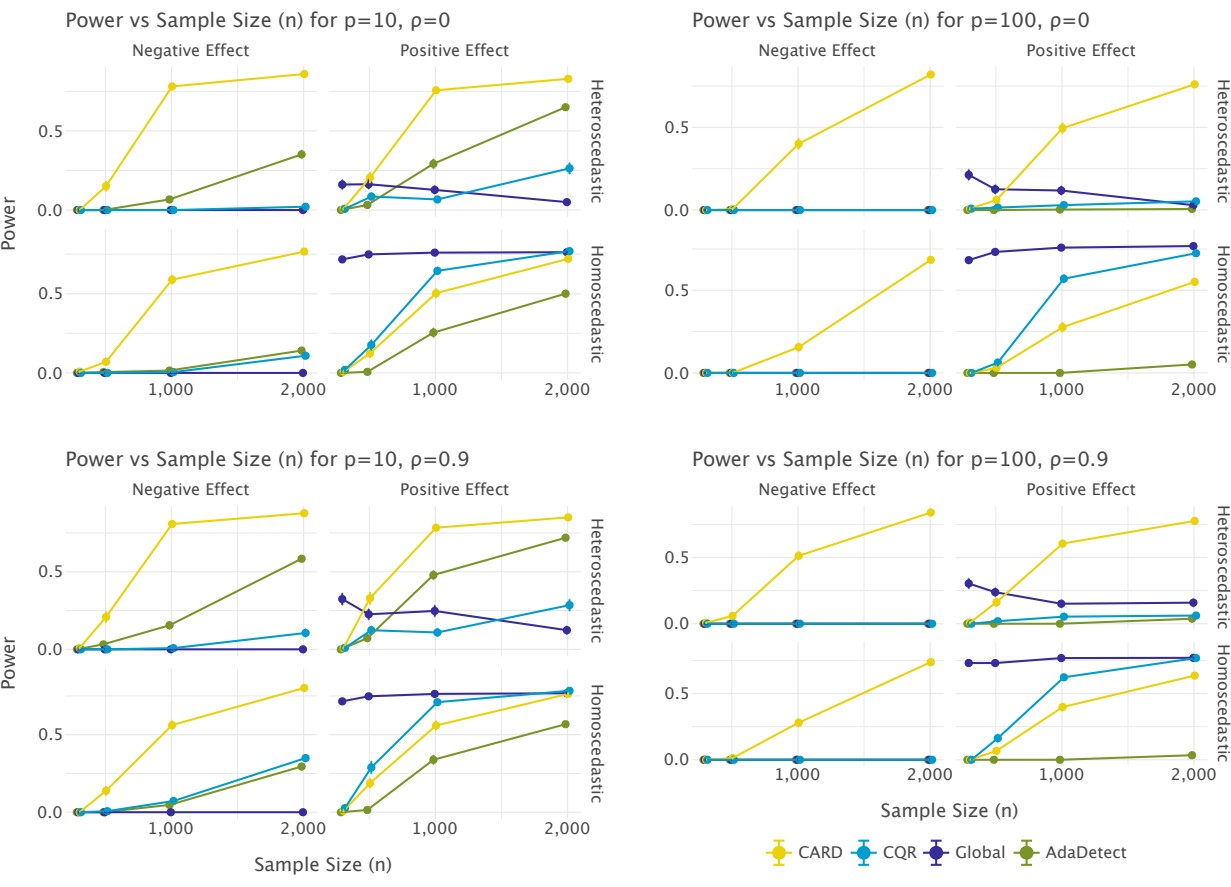

Figure 5: Power analysis of various methods for different sample sizes. The plots display the power as a function of sample size (n) in the various scenarios. Due to the correlations the AdaDetect method performs better compared to its performance when the covariates are independent from one another, but still the best performing method is CARD except in the positive signal and homoscedastic noise, where the CQR and Global methods achieve better power for $n < 2000$.

## D.2   FDR Results

Across all simulations, all methods control the FDR at the expected level of 0.1.

## E   ACIC 2016 Example

We present an illustrative example of an analysis performed using CARD, utilizing the first dataset from the ACIC 2016 challenge (available in the following repository and described in Dorie et al. (2019)). This dataset comprises 4,802 observations and 58 covariates. After applying one-hot encoding to the categorical variables, we retained 82 covariates in total.

The dataset includes information on the two potential outcomes and the treatment assignment for each observation. We analyzed both the full dataset and a sub-sample consisting of 1,000 observations. In our analysis, we applied CARD with propensity adjustment to the original treatment allocation, treating it as an observational study. To simulate a RCT, we reassigned treatment by sampling from a $Ber(0.5)$ distribution. In the RCT setting we applied CARD without a propensity adjustment. In the original treatment allocation, 858 observations were treated, while in the simulated RCT, 2,368 received treatment.

CARD was applied with 20% of the untreated samples serving as knockoffs. To control the FDR, we applied the BH procedure, setting the FDR threshold at 0.1. When necessary, propensity scores were estimated using a random forest classifier.

We report the proportion rejected hypotheses in each setting (Table 1). Surprisingly, CARD seems to perform better in the observational study setting. This is likely due to CARD benefiting from a large untreated sample, where a large number of knockoffs can be sampled, without compromising the scorer performance. This can also be seen in Figure 7, where the p-values in the observational setting are smaller due to the larger number of knockoffs used.

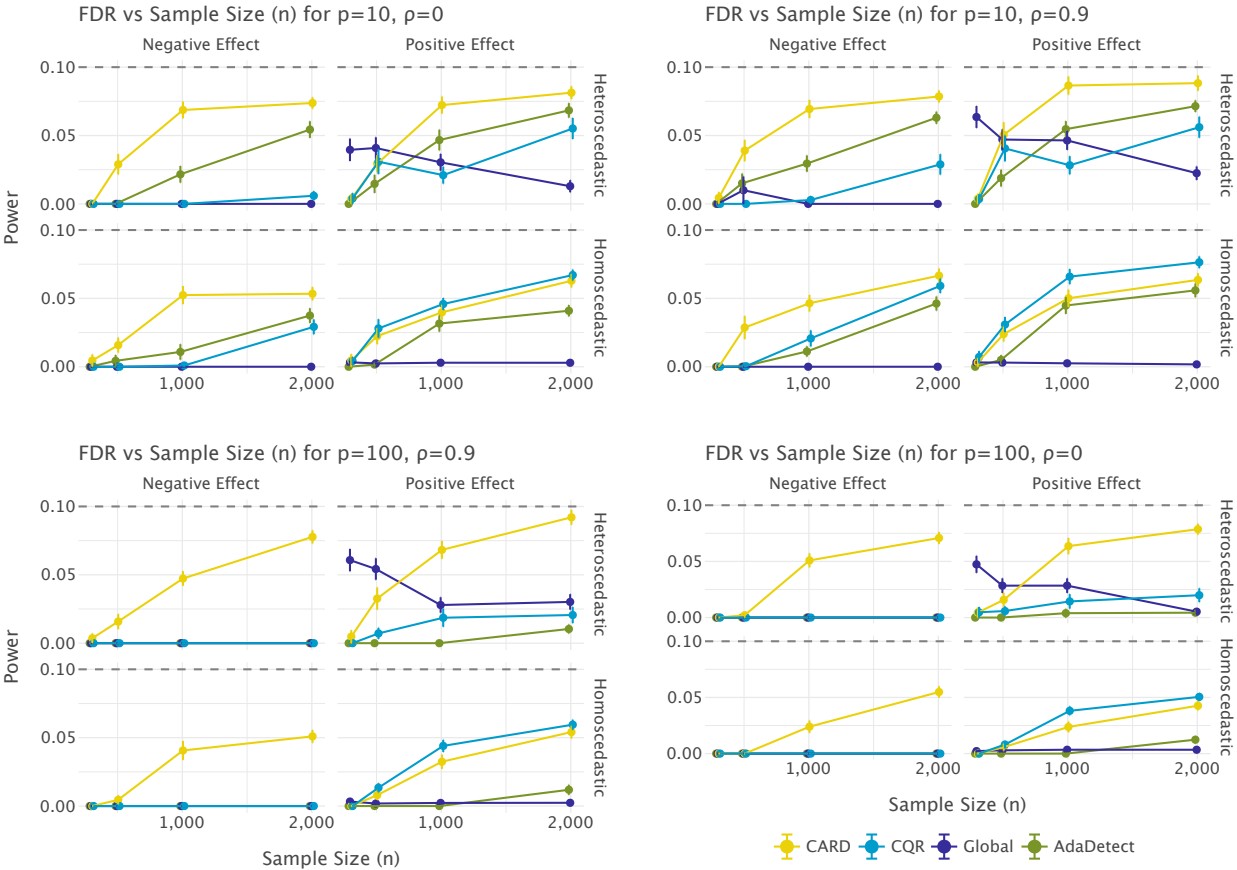

Figure 6: FDR of various methods for different sample sizes. The plots display the FDR as a function of sample size (n).

| $n$ | RCT | Observational Study |
|---|---|---|
| 1000 | 0.53 | 0.73 |
| 4802 | 0.62 | 0.74 |

Table 1: Proportion of rejected hypotheses for the different scenarios for the ACIC 2016 data.

To further analyze the data, we tested for differences between responders and non-responders across each covariate. A Welch's t-test was employed, and we applied a Bonferroni correction (chosen for simplicity in this exploratory post-hoc analysis) to account for multiple comparisons. This analysis was conducted solely under the full RCT setting. The results revealed significant differences in several covariates, most notably for covariate X20, with a p-value of $10^{-72}$. More sophisticated methods, such as logistic regression, can be employed to account for dependencies between covariates, and give further insight on the factors driving the response.

## F    Exploratory Real-World Data Example: Semaglutide in Obstructive Sleep Apnea Patients

We conclude with an illustrative real-world application of CARD to an observational dataset of patients diagnosed with obstructive sleep apnea (OSA). The purpose of this analysis is to demonstrate the empirical behavior of CARD in a high-dimensional, confounded clinical setting, rather than to provide confirmatory inference. In particular, this example highlights how CARD can be used as a discovery-oriented tool for identifying treatment-associated distributional signals in real-world data.

The dataset consists of $23,999$ patients with a diagnosis of OSA and complete baseline covariate information. Patients were followed for two years: a one-year screening period followed by a one-year weight-loss tracking period. Treatment corresponds to the initiation of semaglutide, while the control group comprises patients receiving no pharmacological weight-loss intervention. For the untreated cohort, a weight-loss start date was assigned randomly to avoid immortal time bias. The outcome of interest is the relative weight change from the start of the tracking period to the follow-up.

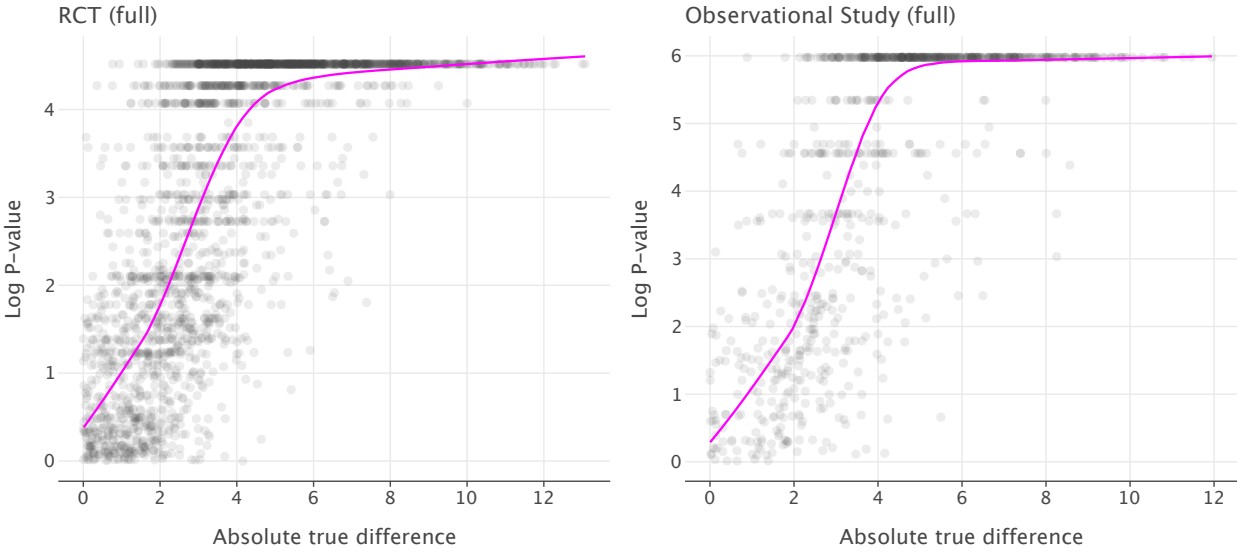

Figure 7: Log-p-values as a function of the true difference, measured by the Conditional Average Treatment Effect (CATE), for the ACIC 2016 dataset. A smooth LOWESS curve (shown in purple) illustrates the trend. As CATE increases, the p-values decrease as expected, with smaller p-values observed in the observational setting, due to the higher number of knockoffs used.

Table 2: CARD diagnostics and exploratory empirical results for the semaglutide cohort. The formal BH-adjusted CARD procedure yielded no FDR-controlled discoveries; the reported responder statistics are based on nominal exploratory p-values.

| Category | Metric | Value |
|---|---|---|
| **Model Diagnostics** | Propensity Score ROC AUC | 0.851 |
| | Common Support (Overlap) [a] | 51.3% |
| | Placebo Discovery Rate [b] | 3.1% |
| **Responder Statistics** | Total Treated Population ($N$) | 2,530 |
| | Identified Exploratory Responders ($n$) | 585 |
| | Proportion of Cohort | 23.1% |
| **Clinical Outcomes** | Mean Weight Change (Responders) | $-8.04\%$ |
| | Mean Weight Change (Non-Responders) | $-2.67\%$ |
| | Standardized Mean Difference (SMD) | $-0.931$ |
| **Top Predictors** | 1. Prior Anti-Diabetic Meds | (Imp: 0.28) |
| | 2. Age at Enrollment | (Imp: 0.16) |
| | 3. Baseline Year | (Imp: 0.11) |

[a] Percentage of patients with propensity scores $e(x) \in [0.05, 0.95]$.
[b] Discovery rate when CARD is applied to a randomly partitioned untreated population (null setting) using $\alpha = 0.1$.

Because treatment assignment is non-random, we estimated the propensity score $e(x) = \mathbb{P}(T = 1 \mid X = x)$ using a random forest classifier with 5-fold cross-fitting. The model achieved strong discrimination (ROC AUC $= 0.851$), indicating substantial selection into treatment based on prior anti-diabetic medication use, age, and baseline BMI. While this strong selection limited the common support to 51.3% of the cohort (see Table 2), the remaining overlap was still sufficient to illustrate the behavior of propensity-adjusted CARD in a challenging real-world setting.

We applied CARD using the responder forest as the non-conformity scorer. Under formal FDR control using the BH procedure at $\alpha = 0.1$, CARD yielded no discoveries. This lack of power likely reflects the severe confounding (propensity $AUC = 0.851$) and limited overlap (51.3%).

To explore whether CARD's scores nonetheless capture treatment-related signal, we conducted an exploratory analysis using nominal (unadjusted) p-values at the same threshold. To validate these exploratory results, we conducted a placebo test by applying CARD to the untreated population with randomized treatment labels (using the same parameters as in the main analysis). This null analysis yielded a discovery rate of only 3.1%, whereas the true treated group yielded a 23.1% responder rate, although it is important to keep in mind that in the placebo case the distribution of covariates of the treated and untreated are similar. The large difference in responders rates suggests that CARD is identifying a treatment-specific signal and not only noise.

Using this nominal threshold, CARD identified 585 responders among $2,530$ treated patients. These responders exhibited a mean weight reduction of $-8.04\%$, compared to only $-2.67\%$ among non-responders. The magnitude of this separation is reflected in a large standardized mean difference (SMD $= -0.931$). The magnitude of this separation indicates that the nominal CARD score is associated with clinically relevant weight-loss differences, while remaining hypothesis-generating.

The overlap-sensitivity analysis in Appendix H suggests that weak overlap mainly reduces power by increasing the variance of the propensity-weighted rank statistic, rather than necessarily inflating FDR. This helps explain why the BH-adjusted procedure yielded no formal discoveries despite nominal CARD scores showing treatment-associated separation.

To further characterize the responder phenotype, we trained a secondary random forest classifier to discriminate between CARD-identified responders and non-responders using baseline covariates. Permutation importance scores identified prior anti-diabetic medication use, younger age, and higher baseline BMI as the primary drivers of treatment response. These exploratory predictors are clinically plausible and broadly consistent with recent real-world evidence on heterogeneous weight loss after GLP-1 receptor agonist initiation (Gasoyan et al., 2024), but they should not be interpreted as validated effect modifiers.

Overall, this example illustrates both the potential and the limitations of CARD in severely confounded real-world settings. While CARD's scores appear to capture treatment-related signal, the inability to achieve formal discoveries under BH correction underscores the importance of sufficient overlap and moderate confounding for achieving sufficient power. By providing a clear separation between responders and non-responders that is robust to placebo-style testing, CARD allows researchers to prioritize specific subgroups for downstream confirmatory studies.

## G   Ablation study: design and findings

We use the ablation study to isolate which components of CARD drive its empirical performance, addressing whether the gains come from the responder-specific scoring rule, the tree/forest design, the knockoff construction, or other implementation choices. The ablation study uses the same DGP family used in our main RCT simulations (Section 4.1), restricted to one scenario so that the varied component of each ablation is the only source of variation in the result. The parameters used are $n = 1000$ observations, $p = 10$ covariates, independent $X$ ($\rho = 0$), heteroscedastic noise $\sigma(x) = -2\ln(x)$, positive predictive signal $r = +1$, and a 0.5-Bernoulli treatment assignment. The responder region is $X_1 > 0.5$, $X_2 > 0.5$. Each ablation cell is averaged over 100 independent replications; error bars are $\pm 1$ s.e.

**Varied components.**   We isolate four design choices that govern CARD's behaviour:

- **Scorer.** The responder-specific forest is compared with a single responder tree and three generic $(X, Y)$ classifiers (random forest, logistic regression, XGBoost). The aim is to attribute CARD's power gain to the responder-specific design rather than to the use of a forest classifier per se.

- **Number of estimators.** The responder forest's `n_estimators` ranges over $\{10, 25, 50, 100\}$. The single-tree case ($n_{\text{est}} = 1$) is omitted from this panel and discussed separately in the scorer comparison, because a single responder tree produces only two distinct scores per leaf and is treated as a degenerate boundary case.

- **Knockoff proportion.**  The fraction of untreated units reserved as knockoffs ranges over $\{0.05, 0.1, 0.2, 0.3, 0.5\}$. Too few knockoffs limit the resolution of the conformal p-value calibration; too many shrink the discriminator's training set.

- **Tree depth.** Within the forest variant, the maximum depth ranges over $\{2, 4, 6, 8, 10\}$, crossed with forest sizes $\{10, 50\}$. Shallower trees give coarser leaves but greater stability; deeper trees may overfit.

At $n = 1000$, Figure 8 shows the responder-specific forest controls FDR at the nominal level (FDR $\approx 0.06$) while achieving power $\approx 0.74$. The three generic $(X, Y)$ classifiers (`rf_xy`, `xgb_xy`, `logistic_xy`) also control FDR but recover only 0.0–0.33 of the power, confirming that the responder-specific recursive partitioning on $T \mid Y$ is the principal source of CARD's power gain rather than the use of a forest classifier. A single responder tree (`card_tree`) controls FDR at the nominal level but its two-valued per-leaf score function offers essentially no individual-level ranking and yields zero rejections; this demonstrates that ensembling is necessary, not merely beneficial, for individual-level power.

CARD is robust to the remaining choices over a wide range. Power as a function of forest size rises from 0.74 at n_estimators = 10 to 0.82 at n_estimators = 25 and plateaus at $\approx 0.83$ for n_estimators $\geq 50$, with empirical FDR held at $\approx 0.06$ throughout. The (depth, n_estimators) heat-map is essentially flat: power varies in $[0.75, 0.83]$ across the full grid. The knockoff proportion exhibits a clear interior optimum: power is essentially zero at ko_num = 0.05 (too few knockoffs for the BH cutoff to be informative at $\alpha = 0.1$), rises to its maximum at ko_num = 0.2, and gradually degrades thereafter as the discriminator's training set shrinks. We recommend ko_num $\in [0.1, 0.3]$ and n_estimators $\geq 25$ as practical defaults. This could vary as the number and ratio between the treated and untreated sample change.

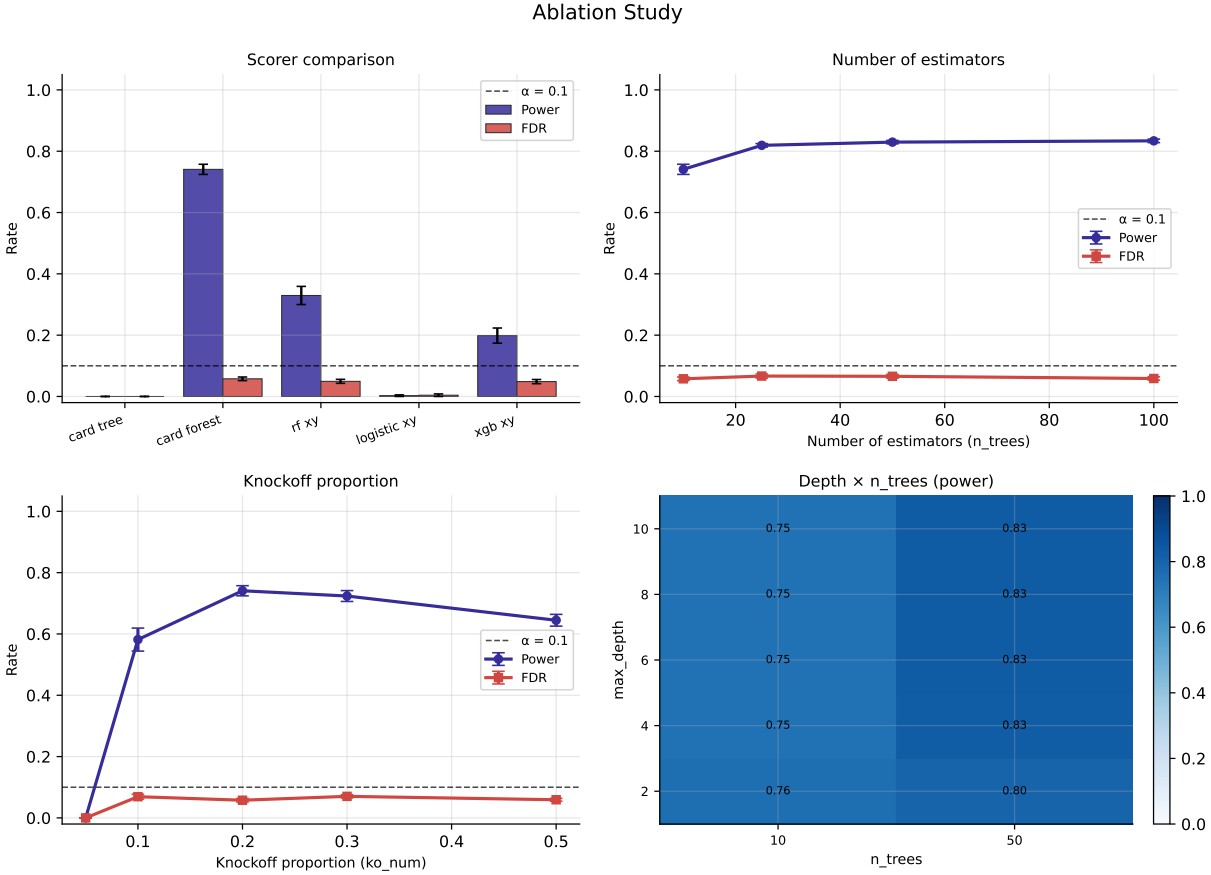

Figure 8: Component ablations on the base RCT scenario ($n = 1000$, $p = 10$, $\rho = 0$, heteroscedastic noise, positive signal). All p-values use a randomized tie-breaking correction (see text). Dashed line marks the nominal $\alpha = 0.1$; bars show $\pm 1$ s.e. over 100 replications. **Top-left:** scorer comparison (`n_estimators`$= 1$ for `card_tree`, $= 10$ for `card_forest`, $= 100$ for generic classifiers). **Top-right:** responder-forest power and empirical FDR as a function of the number of estimators (`n_estimators`$\geq 10$). **Bottom-left:** sensitivity to the knockoff proportion. **Bottom-right:** heat-map of power across tree depth and forest size.

## H    Observational overlap sensitivity

We study sensitivity to overlap on a family of observational DGPs that share a common outcome structure and differ only in how the propensity depends on the covariates. Each replication draws $n = 1000$ units with $p = 10$ uniform-marginal covariates $X \in [0, 1]^{10}$ independent across coordinates, samples treatment $T \sim \mathrm{Bern}(e(X))$, and generates $Y = m(X) + T\, g(X) + \eta$, where the responder boost $g(X) = f(X_0)\, f(X_2)\, \mathbf{1}\{X_0 > 0.5 \cap X_2 > 0.5\}$ for $f(v) = 6/(1 + \exp(-12(v - 0.5)))$ rewards the upper-quadrant of two propensity covariates, and the noise is heteroscedastic in $X_0$ with $\sigma(X_0) = -2\log X_0$. The prognostic $m(X) = 4X_0 + 4X_2 + 4X_5 + 4X_9$ uses every propensity-driving covariate; in the amplified-prognostic variant the three propensity covariates have coefficient 12 instead of 4. A slope $c$ controls overlap. We consider three propensity families: an *additive* log-odds $e(x) = \sigma\big(c(X_0 + X_2 + X_5)/3 - c/2\big)$, a *multiplicative-interaction* log-odds $e(x) = \sigma\big(c(X_0 X_2 X_5 - 1/8)\big)$, and the same multiplicative-interaction propensity paired with the amplified prognostic. Estimated propensities entering the weights are clipped at the default tester threshold of $10^{-3}$. We summarise overlap by the effective sample size ratio under oracle IPW, $\mathrm{ESS}/n = (\sum_i w_i)^2/(n\sum_i w_i^2)$ with $w_i = T_i/e(x_i) + (1 - T_i)/(1 - e(x_i))$, which lies in $(0, 1]$, equals one when $e(x) \equiv 1/2$, and decreases as overlap weakens. The sweep covers $c \in \{0, 0.5, 1, 1.75, 3, 5, 7, 10, 15\}$ in the additive setting and $c \in \{0, 0.5, 4, 8, 12, 20, 30\}$ in the two multiplicative-interaction settings.

We evaluate four propensity-estimation strategies combined with CARD's weighted p-value: the oracle propensity, a correctly-specified additive logistic regression, a random forest with depths $\{3, 10, \infty\}$, and an unadjusted negative control that disables the weighting (`weight_knockoffs = False`). The oracle and the additive logistic recover the true propensity in the univariate and additive multi-dim DGPs but cannot recover the multiplicative interaction $X_0 X_2 X_5$; the random forest cannot recover any of the surfaces exactly in finite samples but bounds its leaf-level predictions away from 0 and 1 by the leaf-averaging construction, with shallower trees trading more fidelity for tighter weights. The unadjusted variant is used as a sanity check: it collapses to the rank comparison between treated and knockoff scores and ignores the fitted propensity.

Figure 9 reports power and FDR versus $\text{ESS}/n$ for the three multi-dim DGPs and the four propensity strategies (RF at depth 3 for the headline figure; the depth sweep is in Figure 10). At $c = 0$ the propensity is $1/2$ everywhere, the weights are uniform, and all four methods coincide at empirical power $\approx 0.96$–0.99 and FDR $\approx 0.07$–0.08 in every DGP; the procedures are observationally indistinguishable in this RCT-like limit. As $c$ grows the oracle and the logistic regression collapse to zero rejections, because their weights become extreme and the variance of the weighted-rank statistic destroys the effective sample. In the additive DGP this collapse occurs over $c \in [3, 5]$ (oracle drops from 0.87 at $c = 3$ to 0.03 at $c = 5$); in the two interaction DGPs the collapse is already complete by $c = 8$ (oracle $\leq 0.01$, logistic $\leq 0.02$). The shallow random forest is the only estimator that retains non-trivial power across the stress range: in the additive DGP it holds power $\approx 0.99$ down to $\text{ESS}/n \approx 0.58$ and degrades smoothly to 0.28 at $\text{ESS}/n \approx 0.39$ before crashing at the very tail; in the interaction DGPs it holds 0.89–0.99 throughout. Empirical FDR for shallow RF stays at or below $\alpha = 0.1$ in the additive DGP, but rises to 0.19 at $\text{ESS}/n \approx 0.18$ under the base-prognostic interaction and to 0.26 at the same overlap level under the amplified-prognostic interaction — both more than two standard errors above $\alpha$ at 100 replications. The negative control fails uniformly: unadjusted FDR rises monotonically to 0.28–0.36 at the smallest $\text{ESS}/n$ across the three DGPs, confirming that the weighting step provides substantial suppression.

Figure 10 resolves the bottom row of Figure 9 into the three random-forest depths $\{3, 10, \infty\}$. Under the multiplicative-interaction with amplified prognostic, deeper random forests (depth 10 and unrestricted) behave essentially like the oracle and logistic estimators: they retain power $\approx 0.7$–0.74 at the highest $\text{ESS}/n$ but collapse to near-zero rejections once $\text{ESS}/n < 0.7$, because their less-regularized predictions produce extreme weights once overlap weakens. Only the depth-3 forest retains power throughout the stress range (power $\approx 0.89$–0.96 across $c \in [4, 30]$). Shallow regularization, not the random-forest class per se, is what preserves power under weak overlap; the same regularisation prevents the estimator from recovering the non-additive propensity surface and so eventually the method fails to control the FDR.

Three findings carry the practical implications for use of CARD in observational settings. First, propensity adjustment is required: the unadjusted negative control breaches $\alpha$ at every overlap regime tested and degrades sharply as overlap weakens, so CARD's asymptotic FDR guarantee in the observational regime relies on the weighting step. Second, an estimator that recovers the propensity surface — oracle or correctly-specified logistic — collapses to zero rejections once $\text{ESS}/n$ drops below roughly 0.8, because the weighted-rank statistic inherits the variance explosion of IPW under poor overlap (**??**). Third, a regularized estimator such as a shallow random forest retains power under weak overlap by bounding its weights, but the same regularization prevents it from recovering a non-additive propensity surface, and the residual bias drives empirical FDP above the nominal level under multiplicative-interaction propensity. A practical recommendation: alongside CARD, check propensity-model adequacy (for instance by cross-validated comparison of an additive parametric fit against a flexible non-parametric alternative) before relying on the FDR guarantee in observational settings with weak overlap.

## I Knockoff randomness and derandomisation

CARD draws its knockoff set randomly from the controls, so two runs on the same dataset with different RNG seeds produce different p-values and different rejection sets. Conditional on the knockoff seed, the BH step is deterministic. We quantify the resulting variability on the four base RCT scenarios ($n = 1000$, $p = 10$, $\rho = 0$): homoscedastic noise with positive predictive effect, homoscedastic with negative predictive effect,

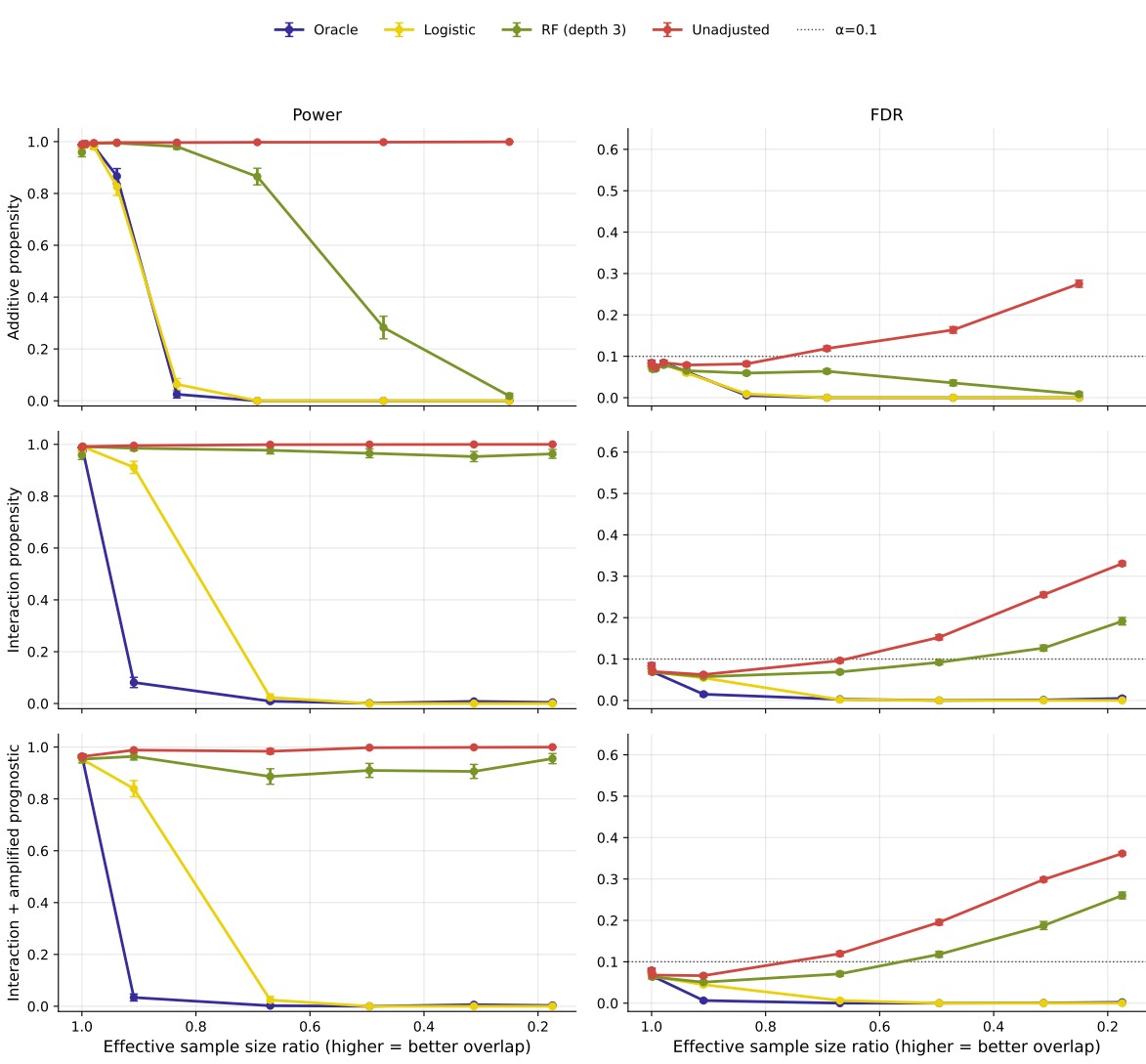

Figure 9: Multi-dim overlap sensitivity. Rows correspond to the three multi-dim DGPs (additive, multiplicative-interaction with base prognostic, multiplicative-interaction with amplified prognostic); columns report empirical power (left) and FDR (right) versus ESS/$n$ for four propensity-estimation strategies (oracle, logistic, shallow RF, unadjusted). True propensity unclipped; estimator output clipped at $10^{-3}$. Dashed line: $\alpha = 0.1$. Error bars are $\pm 1$ s.e. over 100 replications.

heteroscedastic with positive, and heteroscedastic with negative. For each scenario we run 100 replications, each drawing a fresh dataset and a fresh knockoff seed.

Two derandomisation schemes are natural candidates here. The first is the *stability selection* of Meinshausen & Bühlmann (2010): run any selection procedure on $K$ resampled versions of the data and retain only the hypotheses that appear in at least $\lceil \tau K \rceil$ of the $K$ selection sets. Applied to CARD, we draw $K = 20$ independent knockoff resamplings, apply BH at the target level $\alpha$ to each, and form the final rejection set by the majority-vote rule ($\tau = 0.5$). The per-resampling BH step inherits CARD's finite-sample FDR guarantee, and the aggregation preserves it empirically while reducing seed-to-seed variability.

The second candidate is the conformal e-value derandomisation of Bashari et al. (2024): for each of the $K$ resamplings, apply BH at a stricter internal level $\alpha_{\text{int}} = \alpha/10$ to obtain a per-run score threshold $t^{(k)}$,

Figure 10: Random-forest depth sweep under the multiplicative-interaction propensity with amplified prognostic (the bottom-row DGP of Figure 9). Power (left) and FDP (right) vs ESS/$n$ for oracle, logistic, three RF depths $\{3, 10, \infty\}$, and unadjusted as the negative control. Only the depth-3 forest retains non-trivial power across the stress range; depth 10 and unrestricted RF behave like oracle and logistic. FDP for the depth-3 variant rises with $c$ to 0.26 at the smallest ESS/$n$. Error bars are $\pm 1$ s.e. over 100 replications.

construct conformal e-values $e_j^{(k)} = (1 + n_{\mathrm{ko}}) \, \mathbf{1}\{S_j^{(k)} \geq t^{(k)}\}/(1 + \#\{i \text{ knockoff} : S_i^{(k)} \geq t^{(k)}\})$ for each treated unit $j$, average the e-values uniformly across the $K$ resamplings, and apply the e-BH procedure (Wang & Ramdas, 2022) at level $\alpha$ to the aggregated e-values. This method requires that the internal BH step at $\alpha/10$ actually fire on each resampling. For BH at level $\alpha_{\mathrm{int}}$ on $n_{\mathrm{treated}}$ hypotheses to reject anything, the smallest p-value must satisfy $p_{\min} \leq \alpha_{\mathrm{int}}/n_{\mathrm{treated}}$; in CARD the minimum achievable p-value is bounded below by $1/(n_{\mathrm{ko}} + 1)$, so the method requires $n_{\mathrm{ko}} \geq n_{\mathrm{treated}}/\alpha_{\mathrm{int}}$. With the default knockoff fraction of 0.2 on $n_{\mathrm{controls}} = 500$ controls we have $n_{\mathrm{ko}} \approx 80$, while $n_{\mathrm{treated}}/\alpha_{\mathrm{int}} \approx 5 \times 10^4$: the internal BH step is two to three orders of magnitude below the fire threshold and the e-value method produces all-zero e-values per resampling, so e-BH never rejects. This is an inherent limitation of the e-value derandomisation in the knockoff regime — the construction was designed for conformal prediction with large calibration sets, not for knockoff procedures with moderate knockoff counts. We therefore restrict the empirical comparison to single CARD versus stability selection.

Figure 11 shows the power (left) and FDR (right) distributions across the 100 replications for each (scenario, method) cell. Single CARD has substantial seed-to-seed variance: its empirical standard deviation of power ranges from 0.166 to 0.294 across scenarios, i.e. roughly 22–56 % of the mean, and the inter-quartile range spans tens of percentage points. Derandomisation reduces this standard deviation by 2.7–5.6× (to 0.052–0.147) and raises the mean power by 0.05–0.13 relative to single CARD in every scenario. Empirical FDP remains at or below the nominal $\alpha = 0.1$ for both methods, and the derandomised variant has FDR $\approx 0$ throughout.

The seed-to-seed variability of single CARD is large enough that two runs on the same data can differ by twenty percentage points in empirical power. The stability-selection de-randomization of Meinshausen & Bühlmann (2010) both stabilizes and improves CARD: variance shrinks by a factor of three to five, mean power rises by five to thirteen percentage points, and FDR control is preserved across all RCT scenarios. The cost is a $K$-fold increase in compute, with $K = 20$ in our setup. We recommend the de-randomized variant as a practical default whenever the compute budget allows.

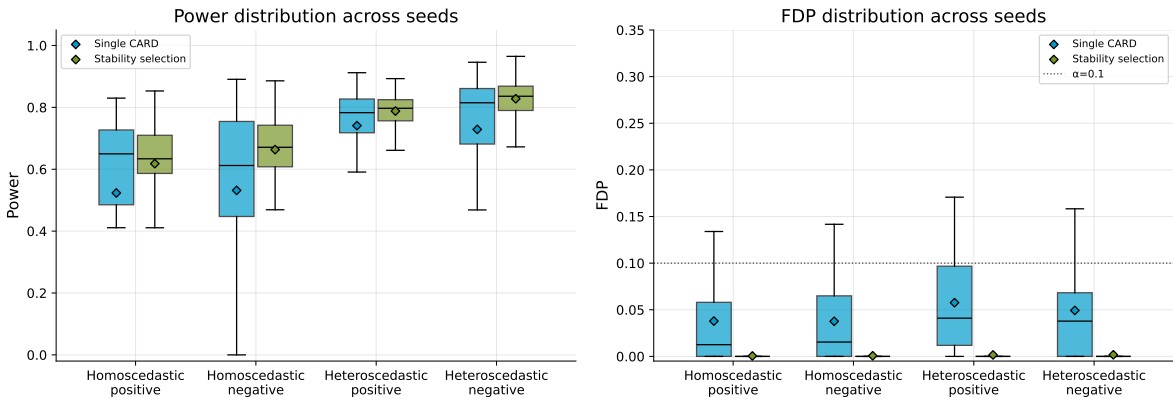

Figure 11: Knockoff-seed randomness and de-randomization. Power (left) and FDR (right) distributions across 100 replications, each drawing a fresh dataset and knockoff seed, for four RCT scenarios ($n = 1000$, $p = 10$, $\rho = 0$). Boxes show the inter-quartile range, whiskers the 5–95 % range, diamonds the mean. Single CARD (cyan) and the stability-selection de-randomization (olive). Dashed line: $\alpha = 0.1$.

