# OpenReview forum: "Detecting Distributional Treatment Responders with False Discovery Rate Control"
_TMLR — Decision pending for TMLR_

### Review · Reviewer_Nj1q · 2026-03-17

**Summary Of Contributions:**

This paper studies responder identification: among treated individuals, identify those whose observed outcomes are statistically inconsistent with the control outcome distribution while controlling the false discovery rate. The proposed method, CARD (causal responder detection), adapts the AdaDetect framework with a responder-specific nonconformity score based on recursive partitioning and further introduces a propensity-weighted version for observational settings. The paper positions this as complementary to CATE analysis: instead of focusing on average treatment effects, it aims to detect general distributional changes such as shifts in variance, skewness, or tails. The main theoretical claim is finite-sample FDR control in randomized experiments and asymptotic FDR control in observational studies under standard causal assumptions. Empirically, the paper presents simulations comparing CARD against global testing, AdaDetect, and CQR, and shows that CARD is often the most powerful method in heterogeneous settings where treatment effects are only detectable after conditioning on covariates.

The main strengths are that the paper tackles an interesting inferential target that is genuinely different from standard CATE estimation, the proposed method is conceptually elegant, and the RCT setting appears technically well motivated.

The main weaknesses are that the observational-study guarantees are materially weaker than the RCT guarantees, the paper only controls type-I error marginally over covariates rather than conditional on covariates, and the empirical evaluation does not yet feel broad enough to fully validate the method’s practical robustness.

**Additional Comments:**

NA

**Audience:**

Yes

**Audience Explanation:**

I believe this paper would be of interest to readers working on causal inference, conformal methods, multiple testing/FDR control, and individualized treatment effect analysis.

**Claims And Evidence:**

No

**Claims Explanation:**

First, the paper’s strongest theoretical support is in the randomized experiment setting, where the conformal/exchangeability argument for FDR control is the clearest. However, for observational studies, the paper’s guarantees are weaker: the submission emphasizes asymptotic FDR control and presents finite-sample simulations suggesting good behavior, but the theoretical support is not on the same footing as in the RCT setting. In particular, the empirical observational study section is limited to CARD variants under a single simulation family and does not fully establish robustness to propensity misspecification or other realistic deviations.

Second, the paper explicitly acknowledges a substantial limitation: the method controls type-I error marginally over treated covariates, not conditional on X. Appendix A shows that conditional type-I error can in fact be arbitrarily bad even while marginal control is preserved. This is an important caveat because responder analysis is often interpreted at the individual or covariate-conditioned level; the current guarantees are weaker than that interpretation may suggest. I think this limitation needs much more prominent treatment in the main paper, and it also tempers how strongly the method can be advertised as identifying responders.

Third, the empirical results are promising but not fully decisive. The simulations show that CARD is often powerful and frequently outperforms AdaDetect and CQR, especially when differences are only detectable after conditioning on covariates. At the same time, the paper also reports regimes where simpler alternatives such as global testing or CQR perform better, especially in the homoscedastic positive-signal setting at smaller sample sizes. That is not fatal, but it means the empirical story is more nuanced than the main narrative sometimes suggests.

**Requested Changes:**

Reframe the main claims for observational studies more carefully. The current presentation risks giving the impression that the observational-setting guarantees are as strong as those in RCTs, whereas the theoretical support is materially weaker. The title, abstract, and introduction should state this distinction more explicitly.

Move the discussion of marginal vs. conditional error control into the main text and make it much more prominent. Appendix A shows this is not a minor technicality: conditional type-I error can be very poor even when marginal guarantees hold. This significantly affects how the method should be interpreted in practice.

Strengthen the empirical evaluation for observational studies. The current experiments are too narrow. The paper should include broader sensitivity analyses covering propensity misspecification, overlap violations, stronger confounding regimes, and more varied data-generating processes.

Expand baseline comparisons or justify more clearly why the current baselines are sufficient. Since the paper targets responder detection rather than standard CATE estimation, it is reasonable not to compare against every ITE estimator, but the paper should more convincingly explain which alternative methods are truly competitive for the stated hypothesis-testing problem.

Add ablations for the responder-specific scorer, including tree depth / forest design / data splitting choices / knockoff proportion, to clarify which components drive the gains over AdaDetect.

Better characterize regimes where CARD is not superior, especially the homoscedastic positive-signal setting where global testing or CQR can outperform it for smaller sample sizes. A more balanced discussion would improve the paper.

Strengthening: Improve exposition around the scientific meaning of “responder.” The paper is strongest when framed as detecting distributional departures from control, and weaker when readers may interpret it as pointwise individualized causal certification.

---

> ### Author Response · Authors · 2026-05-16
>
> We thank the reviewer for the technically detailed evaluation and for clearly identifying the main places where the paper’s claims and empirical support should be strengthened.
> * The reviewer raised six central concerns: observational guarantees, marginal versus conditional control (including relocating the former appendix discussion into the main text), responder interpretation, observational simulations, ablations, and characterization of failure regimes. Each is addressed in the consolidated response, with specific revisions as follows: the observational-guarantee qualifications appear in the revised Abstract, Section 3.3, and Discussion; the marginal-versus-conditional discussion with its illustrative example is now Section 3.4; the responder reframing runs through the title, Abstract, Section 1, Section 1.2, and Discussion; the new observational simulations are in Appendix F; the component ablations are in Appendix E; and the failure-regime discussion appears in Section 4.1 and the Discussion.
> * We agree that the baseline discussion should be strengthened. The current baselines—global testing, AdaDetect, and CQR—were chosen because they produce valid or directly comparable p-values for the distributional  responder-detection null studied in the paper. Standard CATE/ITE estimators target different quantities and are not directly comparable unless embedded in a valid testing procedure. We made this distinction explicit.
> * We also made the empirical narrative more balanced by clarifying CARD’s intended operating regime. CARD is most useful when treatment-related departures are heterogeneous, covariate-dependent, or distributional beyond a simple marginal shift. In simpler homoscedastic settings where responders are already visible from the marginal distribution of Y, global testing or CQR can be more efficient because they avoid the additional cost of learning covariate-adaptive structure. We revised the discussion to present these cases as expected limitations rather than failures of the method.
> * We revised the observational-study claims so that the title, abstract, and introduction do not suggest that observational guarantees are as strong as the RCT guarantees.

---

### Review · Reviewer_8RWg · 2026-04-15

**Summary Of Contributions:**

Strengths:

- Introduces CARD, a novel responder detection method based on distributional differences, not just mean effects.
- Captures variance, skewness, and tail changes, overcoming limitations of CATE-based approaches.
- Provides finite-sample FDR control in RCTs and asymptotic control in observational studies.
- Uses conformal prediction and knockoffs, ensuring robust, distribution-free inference.
- Uses a task-specific responder tree/forest, improving power over generic classifiers.
- Applicable to both randomized and observational data via propensity score adjustment.
- Demonstrates strong empirical power across diverse simulated settings.
- Clearly distinguishes its objective from CATE/ITE-based methods.
- Flexible framework adaptable to subgroup discovery and directional testing.

Limitations:

- “Responder” definition is indirect and does not guarantee a true individual treatment effect.
- Provides only marginal (not conditional) error control, limiting individual-level interpretability.
- No finite-sample FDR guarantees in observational settings.
- Relies on strong causal assumptions (ignorability, overlap).
- Performance depends on propensity score estimation quality.
- Introduces randomness due to data splitting and knockoffs.
- Can be outperformed in simple settings (e.g., homoscedastic positive signals, small 𝑛).
- Computationally more complex than simpler baseline methods.
- Does not provide effect size estimates, only responder identification.

**Audience:**

Yes

**Audience Explanation:**

The content is certainly relevant for the ML community.

**Broader Impact Concerns:**

- The method targets personalized treatment decisions, raising risks of misclassification of responders, which could lead to inappropriate or unequal treatment allocation.
- Reliance on observational data and propensity scores may introduce or amplify biases due to unmeasured confounding, disproportionately affecting vulnerable groups.

**Claims And Evidence:**

Yes

**Claims Explanation:**

-

**Requested Changes:**

- Clarify the interpretation of “responder”, explicitly distinguishing it from true individual treatment effects (ITE).
- Discuss more clearly the practical implications of marginal vs conditional FDR control.
- Provide deeper analysis of failure cases (e.g., homoscedastic + positive signal scenarios).
- Include real-world experiments in the main text (not only appendix) to strengthen empirical validity. (major)
- Report computational complexity and runtime compared to competing methods.
- Add ablation studies to isolate contributions (e.g., scorer vs conformal vs knockoffs). (major)
- Clarify the choice and tuning of the responder tree/forest (depth, splits, hyperparameters).

---

> ### Author Response · Authors · 2026-05-16
>
> We thank the reviewer for the positive and detailed assessment of CARD’s contributions and for clearly identifying practical improvements.
> * The concerns about responder interpretation, marginal FDR control, observational assumptions, ablations, and failure regimes are addressed in the consolidated response above and in the revised manuscript (we have edited the relevant section specifically the causal inference (1.1) and inferential goal (1.2)).
> * We clarified that CARD outputs FDR-controlled discoveries, p-values, and nonconformity scores, not unit-level effect-size estimates of the ITE.
> * We agree that real-world data examples should be more visible. We moved a condensed version of the semaglutide real-world data analysis into the main text, while retaining full details in the appendix (Section 4.3; Appendix D).
> * We clearly distinguished confirmatory findings from exploratory analyses, especially in observational settings with strong confounding or limited overlap.  In particular, for the semaglutide analysis, we state prominently that the formal BH-adjusted CARD procedure yields no FDR-controlled discoveries and that the nominal discoveries are exploratory.
> * We reported sensitivity to knockoff sampling randomness by repeating CARD over multiple random seeds and summarizing variability in empirical FDR and power.
> * We added a qualitative computational discussion. The main computational cost of CARD is fitting the responder tree/forest on the augmented treated/control/knockoff sample; conformal scoring and BH correction are comparatively minor post-processing steps. Thus, with a tree- or forest-based scorer, CARD is not substantially more computationally demanding than fitting a standard tree/forest classifier, or than applying AdaDetect with a comparable tree/forest scorer. We focused the discussion on the main algorithmic bottlenecks rather than reporting implementation-dependent wall-clock runtimes.
> * We added implementation details specifying the responder forest design, including the number of trees, maximum depth, knockoff proportion, and propensity cross-fitting, and added sensitivity analyses for forest size, depth, and knockoff proportion.
> * We added a broader-impact discussion emphasizing that CARD should not be used as the sole basis for treatment allocation. In observational settings, unmeasured confounding, poor overlap, and biased propensity estimation may produce misleading discoveries or unequal performance across subgroups. We recommend validation, overlap diagnostics, sensitivity analyses, and domain oversight before practical deployment.

---

### Review · Reviewer_tGG9 · 2026-05-02

**Summary Of Contributions:**

This paper focuses on the identification of responders in randomized controlled trials and observational studies, and provides a rigorous causal definition of individuals who truly benefit from treatment within the potential outcomes framework.

**Additional Comments:**

This submission is outside my area of expertise, so I focused my evaluation on the clarity of the paper, the precision of its claims, and whether its conclusions are adequately supported by the evidence presented.

**Audience:**

Yes

**Audience Explanation:**

The paper systematically discusses the key assumptions required for identifying responders under different study designs, as well as the corresponding limits of identifiability, and clarifies the relationship among the average treatment effect, individual benefit, and the proportion of responders. Overall, the topic is important and offers meaningful practical implications.

**Claims And Evidence:**

No

**Claims Explanation:**

Some claims alike identifying responders at the individual level is generally not possible without strong additional assumptions under the potential outcomes framework. Likewise, conclusions for observational studies rely heavily on assumptions such as no unmeasured confounding, which should be stated more explicitly. In addition, interpreting the average treatment effect as the proportion of responders requires extra conditions, such as monotonicity.

**Requested Changes:**

First, the interpretation of responders remains somewhat stronger than what the method formally identifies: CARD detects treated units whose observed outcomes are statistically inconsistent with the control distribution, rather than strictly identifying individual-level causal responders. This distinction should be made more explicit in the title, abstract, and main text. Second, the method provides marginal FDR control over treated units, not conditional error control given covariates, and this limitation deserves more prominent discussion. Third, the theoretical guarantees in observational settings are substantially weaker than in randomized experiments, and the paper would benefit from a clearer account of the required assumptions, the role of estimated propensity scores, and the practical limitations under finite samples or poor overlap. Finally, the empirical section could be strengthened with additional baselines, ablations, and more challenging simulation settings to better characterize both the advantages and the limitations of CARD.

---

> ### Author Response · Authors · 2026-05-16
> **Response tGG9**
>
> * We thank the reviewer for focusing on clarity, precision of claims, and whether the conclusions are supported by the evidence.
> * As discussed in the consolidated response, we have revised the manuscript to avoid any implication that CARD identifies individual-level causal responders in the strict counterfactual sense.
> * We have explicitly defined CARD responders as distributional responders, i.e., treated observations whose outcomes are statistically inconsistent with the control response distribution.
> * We also agree with the reviewer’s point that ATE/CATE should not generally be interpreted as a proportion of responders. ATE/CATE summarize average effects, whereas CARD has a distinct inferential target. We have revised the introduction and discussion to avoid any suggestion that CARD decomposes ATE/CATE into responder proportions or that such a relationship holds without additional assumptions such as monotonicity.
> * Regarding observational studies, the required assumptions already appear in the manuscript, but we agree that they should be presented more prominently. We have made the reliance on ignorability, overlap, and propensity-score estimation clearer in the abstract, introduction, propensity-adjusted CARD section, and discussion.
> * We have also expanded the empirical section with ablations and additional observational simulations, as described in the consolidated response (see comments to manuscript).

---

### Author Response · Authors · 2026-05-16
**General Comments**

We thank the reviewers for their careful and constructive feedback. The reviews converge on four issues: the formal meaning of “responder,” the marginal rather than conditional nature of CARD’s error control, the weaker status of the observational-study guarantees, and the need for broader empirical characterization. We agree with these points. In the revision, we have reframed CARD as detecting distributional responders: treated observations whose observed outcome-covariate pairs are statistically inconsistent with the control response distribution under the CARD null, rather than certifying unit-level individual treatment effects. We have revised the title, abstract, introduction, inferential-goal section, and discussion accordingly. The revised title is: “Detecting Distributional Treatment Responders with False Discovery Rate Control.”

Summary of changes in the revised manuscript.
To make the revision easy to evaluate, we briefly summarize the main changes made in response to the reviews. First, we revised the title, abstract, introduction, inferential-goal section, and discussion to consistently frame CARD as detecting distributional responders, rather than certifying unit-level individual treatment effects (see revised title; Abstract; Introduction; Section 1.2; and Discussion). Second, we moved the marginal-versus-conditional error-control discussion into the main text and added an illustrative example showing that conditional Type I error can be poor even when marginal control is preserved (Section 3.4, “Marginal vs. conditional error control”). Third, we strengthened the distinction between randomized and observational settings, emphasizing that CARD has finite-sample FDR control in RCTs, while the observational extension is asymptotic and depends on ignorability, overlap, and adequate propensity-score estimation (Abstract; Section 3.1; Section 3.3; Discussion). Fourth, we expanded the empirical characterization by adding component ablations, observational overlap/propensity sensitivity analyses, and a derandomization analysis for knockoff sampling (Appendices E, F, and G). Finally, we moved a condensed exploratory semaglutide real-world data example into the main text, added a computational-cost discussion, and added a broader-impact discussion (Section 4.3; Appendix D; Discussion; Broader Impact Statement).

---

> ### Author Response · Authors · 2026-05-16
> **Interpretation of “Responder”**
>
> We agree that the manuscript should more sharply distinguish CARD’s formal inferential target from individual-level causal certification. CARD does not certify individual treatment effects, i.e., it does not prove Yi(1)≠Yi(0), since the counterfactual outcome is unobserved. CARD discoveries provide evidence of distributional departure from the control response distribution for the tested treated population, but they are not unit-level causal certificates. In particular, distributional responder status is neither equivalent to nor a necessary or sufficient condition for proving a nonzero individual treatment effect for a particular unit; individual treatment effects may exist even when Y(1) and Y(0)  have the same marginal distribution. We have revised the manuscript to consistently use "distributional responders" and to remove wording that could suggest otherwise (revised title; Abstract; Section 1 paragraph 2; Section 1.2; and the Remark in Section 1.2). We have also clarified that CARD does not rely on interpreting ATE/CATE as a responder proportion; ATE/CATE summarize mean effects, whereas CARD targets distributional inconsistency with the control response distribution (Section 1, paragraph 2; Discussion, paragraph 1).

---

> ### Author Response · Authors · 2026-05-16
> **Marginal vs Conditional Error Control**
>
> * We agree that the marginal nature of the FDR guarantee should be more prominent.
> * CARD controls FDR marginally over the treated units being tested, not conditionally at each fixed covariate value X=x.
> * Thus, discoveries should be interpreted as FDR-controlled over the tested treated population as a whole, not as conditionally valid individualized tests for every patient subgroup or covariate profile.
> * Rejection for a treated unit supports that the unit is statistically inconsistent with the control response distribution under the marginal CARD null. It does not imply that the same error guarantee holds conditionally for all patients with the same covariates.
> * We have moved this discussion into the main text and added an illustrative example showing that conditional Type I error can be poor even when marginal control is preserved.

---

> ### Author Response · Authors · 2026-05-16
> **Randomized vs Observational Guarantees**
>
> * We agree that the manuscript should more clearly distinguish the RCT and observational settings.
> * In randomized experiments, CARD relies on exchangeability and provides finite-sample FDR control.
> * In observational studies, CARD relies on propensity-score adjustment and standard causal assumptions. As such, these guarantees are weaker and are not on the same theoretical footing as the randomized setting.
> * The relevant assumptions already appear in the manuscript, especially in the propensity-adjusted CARD section and Lemma 1, but we agree they should be emphasized earlier and more prominently.
> * We have revised the abstract, introduction, propensity-adjusted CARD section, and discussion to highlight the roles of:
> - ignorability / no unmeasured confounding
> - overlap / positivity
> - propensity-score estimation
> - finite-sample limitations in observational studies.
> * We have explicitly stated that the finite-sample FDR guarantee available in RCTs does not directly carry over to observational studies with estimated propensity scores.

---

> ### Author Response · Authors · 2026-05-16
> **Empirical Characterization**
>
> * We agree that the empirical section should better characterize both the advantages and limitations of CARD.
> * We added observational sensitivity analyses varying overlap strength and propensity structure, comparing oracle propensity, logistic-regression propensity estimation, random-forest propensity estimation, and an unadjusted negative control. These experiments clarify where the observational extension is informative and where finite-sample performance degrades due to weak overlap or propensity-estimation error.
> * We also added ablations isolating major CARD components, including the responder-specific scorer versus generic classifiers, single tree versus forest scoring, knockoff proportion, tree depth, and number of trees (Appendix E). In observational settings, the overlap sensitivity study also compares propensity-adjusted CARD against an unadjusted negative control (Appendix F).
> * Finally, we better characterize regimes where CARD is not expected to dominate. In particular, when the treatment effect is already visible in the marginal distribution of Y, such as homoscedastic positive-signal settings at smaller sample sizes, simpler methods such as global testing or CQR can outperform CARD.

---

### Decision · Action_Editor_eQeC · 2026-07-08

**Recommendation:** Accept as is

**Audience:**

Yes

**Audience Explanation:**

The paper addresses responder detection with FDR control, a problem of clear
interest to the causal inference, conformal prediction, and multiple-testing
communities. It also connects naturally to personalized-medicine and clinical-trial
applications, so both methodological and applied readers at TMLR would find the
findings relevant.

**Claims And Evidence:**

Yes

**Claims Explanation:**

Yes. After the revision, the claims match what the paper actually proves.
Finite-sample FDR control is claimed only for randomized settings (following
Marandon et al., 2024), while the observational extension is clearly presented as
asymptotic and dependent on ignorability, overlap, and propensity estimation. The
reframing to *distributional* responders (title, abstract, Section 1.2, Remark 1)
drops the earlier overstatement about certifying individual treatment effects,
and the marginal-versus-conditional caveat is now made explicit in Section 3.4.
The experiments also honestly show where CARD does not help (Figures 2 and 3, and
the exploratory semaglutide example), which makes the evidence more convincing.